# Incorporating Bias-aware Margins into Contrastive Loss for Collaborative Filtering

**An Zhang**[†‡]  **Wenchang Ma**[‡]  **Xiang Wang**[§*]  **Tat-Seng Chua**[†‡]
[†]Sea-NExT Joint Lab
[‡]National University of Singapore
[§]University of Science and Technology of China
anzhang@u.nus.edu, e0724290@u.nus.edu, xiangwang1223@gmail.com
dcscts@nus.edu.sg

## Abstract

Collaborative filtering (CF) models easily suffer from popularity bias, which makes recommendation deviate from users' actual preferences. However, most current debiasing strategies are prone to playing a trade-off game between head and tail performance, thus inevitably degrading the overall recommendation accuracy. To reduce the negative impact of popularity bias on CF models, we incorporate Bias-aware margins into Contrastive loss and propose a simple yet effective **BC Loss**, where the margin tailors quantitatively to the bias degree of each user-item interaction. We investigate the geometric interpretation of BC loss, then further visualize and theoretically prove that it simultaneously learns better head and tail representations by encouraging the compactness of similar users/items and enlarging the dispersion of dissimilar users/items. Over eight benchmark datasets, we use BC loss to optimize two high-performing CF models. On various evaluation settings (*i.e.,* imbalanced/balanced, temporal split, fully-observed unbiased, tail/head test evaluations), BC loss outperforms the state-of-the-art debiasing and non-debiasing methods with remarkable improvements. Considering the theoretical guarantee and empirical success of BC loss, we advocate using it not just as a debiasing strategy, but also as a standard loss in recommender models. Codes are available at https://github.com/anzhang314/BC-Loss.

## 1   Introduction

At the core of leading collaborative filtering (CF) models is the learning of high-quality representations of users and items from historical interactions. However, most CF models easily suffer from the popularity bias issue in the interaction data [1, 2, 3, 4]. Specifically, the training data distribution is typically long-tailed, *e.g.,* a few head items occupy most of the interactions, whereas the majority of tail items are unpopular and receive little attention. The CF models built upon the imbalanced data are prone to learn the popularity bias and even amplify it by over-recommending head items and under-recommending tail items. As a result, the popularity bias causes the biased representations with poor generalization ability, making recommendations deviate from users' actual preferences.

Motivated by concerns of popularity bias, studies on debiasing have been conducted to lift the tail performance. Unfortunately, most prevalent debiasing strategies focus on the trade-off between head and tail evaluations (see Table 3), including post-processing re-ranking [5, 6, 7, 8, 9], balanced training loss [10, 11, 12, 9], sample re-weighting [13, 14, 15, 16, 17, 18], and head bias removal by causal inference [19, 20, 21, 22]. Worse still, many of them hold some assumptions that are

---

[*]Xiang Wang is the corresponding author.

36th Conference on Neural Information Processing Systems (NeurIPS 2022).

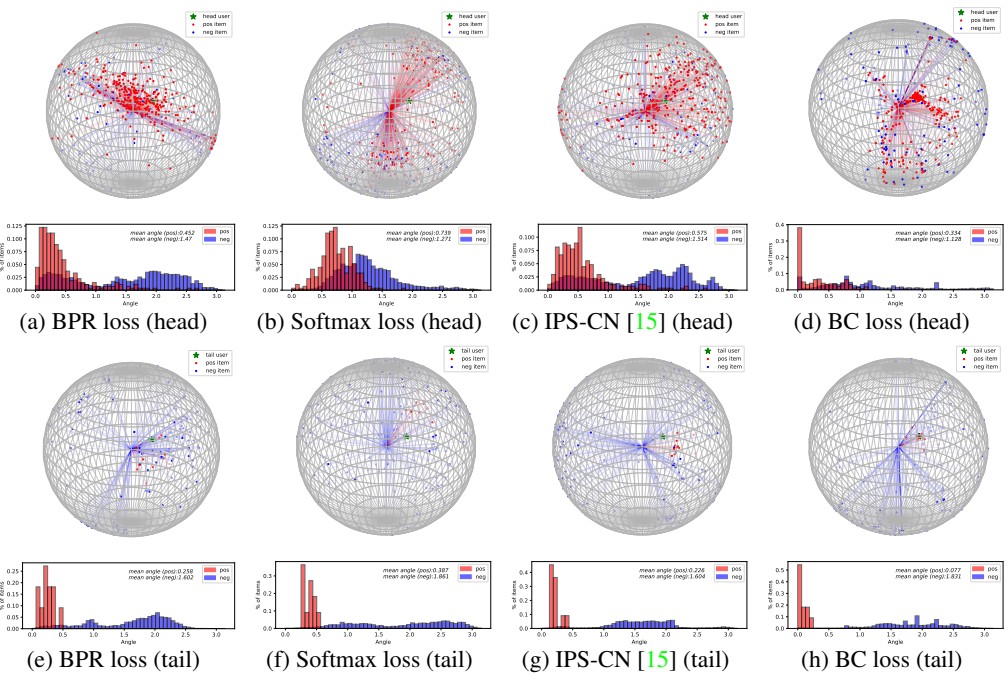

Figure 1: Visualizations of item representations learned by LightGCN [25] on Yelp2018 [25], where subfigures (a-d)/(e-h) depict the identical head/tail user as a green star, while the red and blue points denote positive and negative items, respectively. In each subfigure, the first row presents the 3D item representations projected on the unit sphere, while the second row shows the angle distribution of items *w.r.t.* the specific user and the statistics of mean angles. Compared to other losses, BC loss learns better head representations (*cf.* with the smallest mean positive angle, the vast majority of positive items fall into the group closest to the user) and tail representations (*cf.* a clear margin exists between positive and negative items for the tail user). BC loss learns a more reasonable representation distribution that is locally clustered and globally separated. See more details in Appendix A.1.

infeasible in practice, such as the balanced test distribution is known in advance to guide the hyperparameters' adjustment [23, 22], or a small unbiased data is present to train the unbiased model [24, 19]. Consequently, they pursue improvements on tail items but exacerbate the performance sacrifice of head items, leading to a severe overall performance drop. The trade-off between the head and tail evaluations results in suboptimal representations, which derails the generalization ability.

In this paper, we conjecture that an ideal debiasing strategy should learn high-quality head and tail representations with powerful discrimination and generalization abilities, rather than playing a trade-off game between the head and tail performance. Here we follow the prior studies [15, 23, 13, 14] to focus on one key ingredient in representation learning: the loss function. Figure 1 depicts the item representations, which is optimized via two non-debiasing losses (BPR [26] and Softmax [27]) and one debiasing loss (IPS-CN [15]). Wherein, representation discrimination is reflected in how well the positive items of a user are apart from the negatives. Our insights are: (1) For a user, the non-debiasing losses are inadequate to discriminate his/her positive and negative items well, since their representations are largely overlapped as Figures 4a and 4b show; (2) Although IPS-CN achieves better discrimination power in the tail group than BPR (*cf.* positive items get smaller angles to the ego user in Figure 4g, as compared to Figure 4e), it gets worse discrimination ability in the head (*cf.* positive items hold larger angles to the ego user in Figure 4c, as compared to Figure 4a).

Towards this end, we incorporate Bias-aware margins into Contrastive Loss and devise a simple yet effective **BC Loss** to guide the head and tail representation learning of CF models. Specifically, we first employ a bias degree extractor to quantify the influence of interaction-wise popularity — that is, how well an interaction is predicted, when only popularity information of the target user and item is used. Interactions involving inactive users and unpopular items often align with lower bias degrees, indicating that popularity fails to reflect user preference faithfully. In contrast, interactions with active users and popular items are spurred by the popularity information, thus easily inclining to

high bias degrees. We then move on to train the CF model by converting the bias degrees into the angular margins between user and item representations. If the bias degree is low, we impose a larger margin to strongly squeeze the tightness of representations. In contrast, if the bias degree is large, we exert a small or vanishing margin to reduce the influences of biased representations. Through this way, for each ego user's representation, BC quantitatively controls its bias-aware margins with item representations — adaptively intensifying the representation similarity among positive items, while diluting that among negative items. Benefiting from stringent and discriminative representations, BC loss significantly improves both head and tail performance.

Furthermore, BC loss has three desirable advantages. First, it has a clear geometric interpretation, as illustrated in Figure 2. Second, it brings forth a simple but effective mechanism of hard example mining (See Appendix A.2). Third, we theoretically reveal that BC loss tends to learn a low-entropy cluster for positive pairs (*e.g.,* compactness of matched users and items) and a high-entropy space for negative pairs (*e.g.,* dispersion of unmatched users and items) (See Theorem 1). Considering the theoretical guarantee and empirical effectiveness, we argue that BC loss is not only promising to alleviate popularity bias, but also suitable as a standard learning strategy in CF.

## 2 Preliminary of Collaborative Filtering (CF)

**Task Formulation.** Personalized recommendation is retrieving a subset of items from a large catalog to match user preference. Here we consider a typical scenario, collaborative filtering (CF) with implicit feedback [28], which can be framed as a top-$N$ recommendation problem. Let $\mathcal{O}^+ = \{(u,i)|y_{ui} = 1\}$ be the historical interactions between users $\mathcal{U}$ and items $\mathcal{I}$, where $y_{ui} = 1$ indicates that user $u \in \mathcal{U}$ has adopted item $i \in \mathcal{I}$ before. Our goal is to optimize a CF model $\hat{y} : \mathbb{U} \times \mathbb{I} \to \mathbb{R}$ that latches on user preference towards items.

**Modeling Scheme.** Scrutinizing leading CF models [26, 25, 29, 30], we systematize the common paradigm as a combination of three modules: user encoder $\psi(\cdot)$, item encoder $\phi(\cdot)$, and similarity function $s(\cdot)$. Formally, we depict one CF model as $\hat{y}(u, i) = s(\psi(u), \phi(i))$, where $\psi : \mathbb{U} \to \mathbb{R}^d$ and $\phi : \mathbb{I} \to \mathbb{R}^d$ encode the identity (ID) information of user $u$ and item $i$ into $d$-dimensional representations, respectively; $s : \mathbb{R}^d \times \mathbb{R}^d \to \mathbb{R}$ measures the similarity between user and item representations. In literature, there are various choices of encoders and similarity functions:

- Common encoders roughly fall into three groups: ID-based (*e.g.,* MF [26, 29], NMF [31], CMN [32]), history-based (*e.g.,* SVD++ [29], FISM [33], MultVAE [30]), and graph-based (*e.g.,* GCMC [34], PinSage [35], LightGCN [25]) fashions. Here we select two high-performing encoders, MF and LightGCN, as the backbone models being optimized.

- The widely-used similarity functions include dot product [26], cosine similarity [36], and neural networks [31]. As suggested in the recent study [36], cosine similarity is a simple yet effective and efficient similarity function in CF models, having achieved strong performance. For better interpretation, we take a geometric view and denote it by:

$$s(\psi(u), \phi(i)) = \frac{\psi(u)^\top \phi(i)}{\|\psi(u)\| \cdot \|\phi(i)\|} \doteq \cos(\hat{\theta}_{ui}), \qquad (1)$$

in which $\hat{\theta}_{ui}$ is the angle between the user representation $\psi(u)$ and item representation $\phi(i)$.

**Learning Strategy.** To optimize the model parameters, CF models mostly frame the top-$N$ recommendation problem into a supervised learning task, and resort to one of three classical learning strategies: pointwise loss (*e.g.,* binary cross-entropy [37], mean square error [29]), pairwise loss (*e.g.,* BPR [26], WARP [38]), and softmax loss [28]. Among them, pointwise and pairwise losses are long-standing and widely-adopted objective functions in CF. However, extensive studies [9, 1, 39] have analytically and empirically confirmed that using pointwise or pairwise loss is prune to propagate more information towards the head user-item pairs, which amplifies popularity bias.

Softmax loss is much less explored in CF than its application in other domains like CV [40, 41]. Recent studies [36, 42, 43, 44, 45] find that it inherently conducts hard example mining over multiple negatives and aligns well with the ranking metric, thus attracting a surge of interest in recommendation.

Hence, we cast the minimization of softmax loss [27] as the representative learning strategy:

$$\mathcal{L}_0 = - \sum_{(u,i)\in\mathcal{O}^+} \log \frac{\exp\left(\cos(\hat{\theta}_{ui})/\tau\right)}{\exp\left(\cos(\hat{\theta}_{ui})/\tau\right) + \sum_{j\in\mathcal{N}_u} \exp\left(\cos(\hat{\theta}_{uj})/\tau\right)}, \tag{2}$$

where $(u, i) \in \mathcal{O}^+$ is one observed interaction of user $u$, while $\mathcal{N}_u = \{j|y_{uj} = 0\}$ is the set of sampled unobserved items that $u$ did not interact with before; $\tau$ is the hyper-parameter known as the temperature in softmax [46]. Nonetheless, modifying softmax loss to enhance the discriminative power of representations and alleviate the popularity bias remains largely unexplored. Therefore, our work aims to devise a more generic and broadly-applicable variant of softmax loss for CF tasks, which can improve the long-tail performance fundamentally.

## 3 Methodology of BC Loss

On the basis of softmax loss, we devise our BC loss and present its desirable characteristics.

### 3.1 Popularity Bias Extractor

Before mitigating popularity bias, we need to quantify the influence of popularity bias on a single user-item pair. One straightforward solution is to compare the performance difference between the biased and unbiased evaluations. However, this is not feasible as the unbiased data is usually unavailable in practice. Statistical metrics of popularity could be a reasonable proxy of the biased information, such as user popularity statistics $p_u \in \mathbb{P}$ (*i.e.,* the number of historical items that user $u$ has interacted with before) and item popularity statistics $p_i \in \mathbb{P}$ (*i.e.,* the number of observed interactions that item $i$ is involved in). If the impact of the interaction between $u$ and $i$ can be captured well based solely on such statistics, the model is susceptible to exploiting popularity bias for prediction. Hence, we argue that the popularity-only prediction will delineate the influence of bias.

Towards this end, we first train an additional module, termed popularity bias extractor, which only takes the popularity statistics as input to make prediction. Similar to the modeling of CF (*cf.* Section 2), the bias extractor is formulated as a function $\hat{y}_b : \mathbb{P} \times \mathbb{P} \to \mathbb{R}$:

$$\hat{y}_b(p_u, p_i) = s(\psi_b(p_u), \phi_b(p_i)) \doteq \cos(\hat{\xi}_{ui}), \tag{3}$$

where the user popularity encoder $\psi_b : \mathbb{P} \to \mathbb{R}^d$ and the item popularity encoder $\phi_b : \mathbb{P} \to \mathbb{R}^d$ map the popularity statistics of user $u$ and item $i$ into $d$-dimensional popularity embeddings $\psi_b(p_u)$ and $\phi_b(p_i)$, respectively; $s : \mathbb{R}^d \times \mathbb{R}^d \to \mathbb{R}$ is the cosine similarity function between popularity embeddings (*cf.* Equation (1)). $\hat{\xi}_{ui}$ is the angle between $\psi_b(p_u)$ and $\phi_b(p_i)$.

We then minimize the following softmax loss to optimize the popularity bias extractor:

$$\mathcal{L}_b = - \sum_{(u,i)\in\mathcal{O}^+} \log \frac{\exp\left(\cos(\hat{\xi}_{ui})/\tau\right)}{\exp\left(\cos(\hat{\xi}_{ui})/\tau\right) + \sum_{j\in\mathcal{N}_u} \exp\left(\cos(\hat{\xi}_{uj})/\tau\right)}. \tag{4}$$

This optimization enforces the extractor to reconstruct the historical interactions using only biased information (*i.e.,* popularity statistics) and makes the reconstruction reflect the interaction-wise bias degree. As shown in Appendix B.5, interactions with active users and popular items are inclining to learn well via Equation (4). Furthermore, we can distinguish hard interactions based on the bias degree, *i.e.,* the interactions that can be hardly predicted by popularity statistics ought to be more informative for representation learning in the target CF model. In a nutshell, the popularity bias extractor underscores the bias degree of each user-item interaction, which substantively reflects how hard it is to be predicted.

### 3.2 BC Loss

We move on to devise a new BC loss for the target CF model. Our BC loss stems from softmax loss but converts the interaction-bias degrees into the bias-aware angular margins among the representations to enhance the discriminative power of representations. Our BC loss is:

$$\mathcal{L}_{\text{BC}} = - \sum_{(u,i)\in\mathcal{O}^+} \log \frac{\exp\left(\cos(\hat{\theta}_{ui} + M_{ui})/\tau\right)}{\exp\left(\cos(\hat{\theta}_{ui} + M_{ui})/\tau\right) + \sum_{j\in\mathcal{N}_u} \exp\left(\cos(\hat{\theta}_{uj})/\tau\right)}, \tag{5}$$

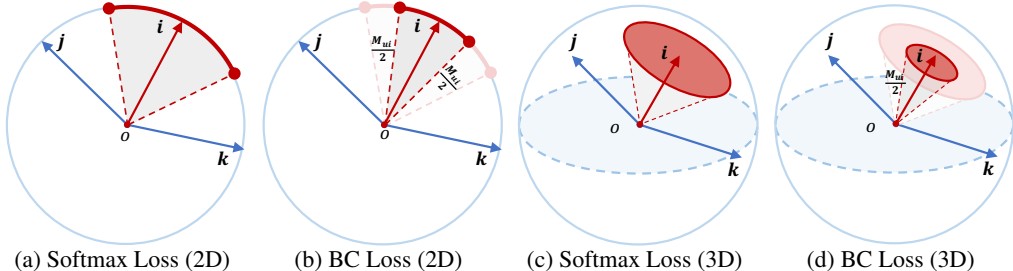

|  |  |  |  |
|:---:|:---:|:---:|:---:|
| (a) Softmax Loss (2D) | (b) BC Loss (2D) | (c) Softmax Loss (3D) | (d) BC Loss (3D) |

Figure 2: Geometric Interpretation of softmax loss and BC loss in 2D and 3D hypersphere. The dark red region indicates the discriminative user constraint, while the light red region is for comparison.

where $M_{ui}$ is the bias-aware angular margin for the interaction $(u, i)$ defined as:

$$M_{ui} = \min\{\hat{\xi}_{ui}, \pi - \hat{\theta}_{ui}\} \tag{6}$$

where $\hat{\xi}_{ui}$ is derived from the popularity bias extractor (*cf.* Equation (3)), and $\pi - \hat{\theta}_{ui}$ is the upper bound to restrict $\cos(\cdot + M_{ui})$ to be a monotonically decreasing function. Intuitively, if a user-item pair $(u, i)$ is the hard interaction that can hardly be reconstructed by its popularity statistics, it holds a high value of $\hat{\xi}_{ui}$ and leads to a high value of $M_{ui}$; henceforward, BC loss imposes the large angular margin $M_{ui}$ between the negative item $j$ and positive item $i$ and optimizes the representations of user $u$ and item $i$ to lower $\hat{\xi}_{ui}$. See more details and analyses in Section 4.

It is noted that BC loss is extremely easy to implement in recommendation tasks, which only needs to revise several lines of code. Moreover, compared with softmax loss, BC loss only adds negligible computational complexity during training (*cf.* Table 5) but achieves more discriminative representations. Hence, we recommend to use BC not only as a debiasing strategy to alleviate the popularity bias, but also as a standard loss in recommender models to enhance the discriminative power. Note that the modeling of $M_{ui}$ is worth exploring, such as the more complex version $M_{ui} = \min\{\lambda \cdot \hat{\xi}_{ui}, \pi - \hat{\theta}_{ui}\}$ where $\lambda$ controls the strength of the bias-margin. Meanwhile, carefully designing a monotonically decreasing function helps to get rid of the upper bound restriction. We will leave the exploration of bias-margin in future work.

## 4 Analyses of BC Loss

We analyze desirable characteristics of BC loss. Specifically, we start by presenting its geometric interpretation, and then show its theoretical properties *w.r.t.* compactness and dispersion of representations. The hard mining mechanism of BC loss is discussed in Appendix A.2.

### 4.1 Geometric Interpretation

Here we probe into the ranking criteria of softmax loss and BC loss, from the geometric perspective. To simplify the geometric interpretation, we analyze one user $u$ with one observed item $i$ and only two unobserved items $j$ and $k$. Then the posterior probabilities obtained by softmax loss are: $\frac{\exp\left(\cos(\hat{\theta}_{ui})/\tau\right)}{\exp\left(\cos(\hat{\theta}_{ui})/\tau\right)+\exp\left(\cos(\hat{\theta}_{uj})/\tau\right)+\exp\left(\cos(\hat{\theta}_{uk})/\tau\right)}$. During training, softmax loss encourages the ranking criteria $\hat{\theta}_{ui} < \hat{\theta}_{uj}$ and $\hat{\theta}_{ui} < \hat{\theta}_{uk}$ to model the basic assumption that the observed interaction $(u, i)$ indicates more positive cues of user preference than the unobserved interactions $(u, j)$ and $(u, k)$.

Intuitively, to make the ranking criteria more stringent, we can impose an angular margin $M_{ui}$ on it and establish a new criteria $\hat{\theta}_{ui} + M_{ui} < \hat{\theta}_{uj}$ and $\hat{\theta}_{ui} + M_{ui} < \hat{\theta}_{uk}$. Directly formulating this idea arrives at the posterior probabilities of BC loss: $\frac{\exp\left(\cos(\hat{\theta}_{ui}+M_{ui})/\tau\right)}{\exp\left(\cos(\hat{\theta}_{ui}+M_{ui})/\tau\right)+\exp\left(\cos(\hat{\theta}_{uj})/\tau\right)+\exp\left(\cos(\hat{\theta}_{uk})/\tau\right)}$. Obviously, BC loss is more rigorous about the ranking assumption compared with softmax loss. See Appendix A.2 for more detailed explanations.

We then depict the geometric interpretation and comparison of softmax loss and BC loss in Figure 2. Assume the learned representations of $i$, $j$, and $k$ are given, and softmax and BC losses are optimized to the same value. In softmax loss, the constraint boundaries for correctly ranking user $u$'s preference

are $\hat{\theta}_{ui} = \hat{\theta}_{uj}$ and $\hat{\theta}_{ui} = \hat{\theta}_{uk}$; whereas, in BC loss, the constraint boundaries are $\hat{\theta}_{ui} + M_{ui} = \hat{\theta}_{uj}$ and $\hat{\theta}_{ui} + M_{ui} = \hat{\theta}_{uk}$. Geometrically, from softmax loss (*cf.* Figure 2c) to BC loss (*cf.* Figure 2d), it is a more stringent circle-like region on the unit sphere in the 3D case. Further enlarging the margin $M_{ui}$ will lead to a smaller hyperspherical cap, which is an explicit discriminative constraint on a manifold. As a result, limited constraint regions squeeze the tightness of similar items and encourages the separation of dissimilar items. Moreover, with the increase of representation dimension, BC loss has more restricted learning requirements, exponentially decreasing the area of constraint regions for correct ranking, and becomes progressively powerful to learn discriminative representations.

## 4.2 Theoretical Properties

BC loss improves head and tail representation learning by enforcing the compactness of matched users and items, while imposing the dispersion of unmatched users and items. See detailed proof in Appendix A.3.

**Theorem 1.** *Let* $\mathbf{v}_u \doteq \psi(u)$, $\mathbf{v}_i \doteq \phi(i)$, *and* $\mathbf{c}_u = \frac{1}{|\mathcal{P}_u|} \sum_{i \in \mathcal{P}_u} \mathbf{v}_i$, $\mathbf{c}_i = \frac{1}{|\mathcal{P}_i|} \sum_{u \in \mathcal{P}_i} \mathbf{v}_u$, *where* $\mathcal{P}_u = \{i | y_{ui} = 1\}$ *and* $\mathcal{N}_u = \{i | y_{ui} = 0\}$ *are the sets of user* $u$*'s positive and negative items, respectively;* $\mathcal{P}_i = \{u | y_{ui} = 1\}$ *is the set of item* $i$*'s positive users. Assuming the representations of users and items are normalized, the minimization of BC loss is equivalent to minimizing a compactness part and a dispersion part simultaneously:*

$$\mathcal{L}_{BC} \geq \underbrace{\sum_{u \in \mathcal{U}} \|\mathbf{v}_u - \mathbf{c}_u\|^2 + \sum_{i \in \mathcal{I}} \|\mathbf{v}_i - \mathbf{c}_i\|^2}_{Compactness\ part} - \underbrace{\sum_{u \in \mathcal{U}} \sum_{j \in \mathcal{N}_u} \|\mathbf{v}_u - \mathbf{v}_j\|^2}_{Dispersion\ part} \propto \underbrace{H(\mathbf{V}|Y)}_{Compactness} \underbrace{-H(\mathbf{V})}_{Dispersion}. \quad (7)$$

**Discussion.** $\mathbf{c}_u$ is the averaged representations of all items that $u$ has interacted with, which describes $u$'s interest; similarly, $\mathbf{c}_i$ profiles item $i$'s user group. For the compactness part, BC loss forces the user's positive items to be user-centric and vice versa. From the entropy perspective, compactness part tends to learn a low-entropy cluster for positive interactions, *i.e.,* high compactness for similar users and items. For the dispersion part, for users and items from unobserved interactions, BC loss maximizes the pairwise euclidean distance between their representations and encourages them to be distant from each other; Hence, from the entropy viewpoint, dispersion part levers the spread of representations to learn a high-entropy representation space, *i.e.,* large separation degree for dissimilar users and items.

## 5 Experiments

We aim to answer the following research questions:

- **RQ1:** How does BC Loss perform compared with debiasing strategies in various evaluations?
- **RQ2:** Does BC loss cause the trade-off between head and tail performance?
- **RQ3:** What are the impacts of the components (*e.g.,* temperature, margin) on BC Loss?

**Baselines & Datasets.** SOTA debiasing strategies in various research lines are compared: sample re-weighting (IPS-CN [15]), bias removal by causal inference (MACR [22], CausE [19]), and regularization-based framework (sam+reg [12]). Extensive experiments are conducted on eight real-world benchmark datasets: Tencent [47], Amazon-Book [48], Alibaba-iFashion [49], Yelp2018 [25], Douban Movie [50], Yahoo!R3 [51], Coat [13] and KuaiRec [52]. For comprehensive comparisons, almost all standard test distributions in CF are covered in the experiments: balanced test set [22, 23, 24], randomly selected imbalanced test set [10, 53], temporal split test set [20, 21, 12], and unbiased test set [13, 52, 51]. See more experiments on KuaiRec, Yahoo!R3, and Coat for unbiased test evaluation in Appendix B.3 and more comparison results between BC loss and other standard losses (most widely used BPR [26], newest proposed CCL [54] and SSM [55]) in Appendix B.4.

### 5.1 Performance Comparison (RQ1)

#### 5.1.1 Evaluations on Imbalanced and Balanced Test Sets

**Motivation.** Many prevalent debiasing methods assume that test distribution is known in advance [22, 23, 10], *i.e.,* the validation set has similar distribution with the test set. Moreover, only an

Table 1: Overall debiasing performance comparison in balanced and imbalanced test sets.

| | Tencent | | | | Amazon-book | | | | Alibaba-iFashion | | | |
| | Balanced | | Imbalanced | | Balanced | | Imbalanced | | Balanced | | Imbalanced | |
| | Recall | NDCG | Recall | NDCG | Recall | NDCG | Recall | NDCG | Recall | NDCG | Recall | NDCG |
|---|---|---|---|---|---|---|---|---|---|---|---|---|
| MostPop | 0.0002 | 0.0002 | 0.0384 | 0.0208 | 0.0001 | 0.0001 | 0.0102 | 0.0063 | 0.0003 | 0.0001 | 0.0212 | 0.0084 |
| MF | 0.0052 | 0.0040 | 0.0982 | 0.0643 | 0.0109 | 0.0103 | 0.0856 | 0.0638 | 0.0056 | 0.0028 | 0.0843 | 0.0411 |
| + IPS-CN | 0.0075 | 0.0058 | 0.0686 | 0.0421 | 0.0132 | 0.0123 | 0.0765 | 0.0554 | 0.0050 | 0.0027 | 0.0551 | 0.0255 |
| + CausE | 0.0056 | 0.0043 | 0.0687 | 0.0468 | 0.0115 | 0.0105 | 0.0720 | 0.0551 | 0.0005 | 0.0003 | 0.0185 | 0.0086 |
| + sam+reg | 0.0070 | 0.0054 | 0.0406 | 0.0266 | 0.0141 | 0.0132 | 0.0599 | 0.0443 | 0.0067 | 0.0032 | 0.0305 | 0.0146 |
| + MACR | 0.0067 | 0.0046 | 0.0326 | 0.0241 | 0.0181 | 0.0146 | 0.0292 | 0.0229 | 0.0086 | 0.0041 | 0.0650 | 0.0331 |
| + BC Loss | **0.0087*** | **0.0068*** | **0.1298*** | **0.0904*** | **0.0221*** | **0.0202*** | **0.1198*** | **0.0948*** | **0.0095*** | **0.0048*** | **0.0967*** | **0.0487*** |
| Imp. % | 16.0% | 17.2% | 32.2% | 40.1% | 22.1% | 38.4% | 40.0% | 49.6% | 10.5% | 17.1% | 14.7% | 18.5% |
| LightGCN | 0.0055 | 0.0042 | 0.1065 | 0.0712 | 0.0123 | 0.0116 | 0.0941 | 0.0724 | 0.0036 | 0.0017 | 0.0660 | 0.0322 |
| + IPS-CN | 0.0072 | 0.0054 | 0.0900 | 0.0599 | 0.0148 | 0.0136 | 0.0836 | 0.0639 | 0.0038 | 0.0017 | 0.0658 | 0.0317 |
| + CausE | 0.0055 | 0.0040 | 0.0966 | 0.0665 | 0.0134 | 0.0121 | 0.0926 | 0.0717 | 0.0029 | 0.0013 | 0.0449 | 0.0221 |
| + sam+reg | 0.0076 | 0.0056 | 0.0653 | 0.0436 | 0.0157 | 0.0149 | 0.0773 | 0.0600 | 0.0056 | 0.0027 | 0.0502 | 0.0252 |
| + MACR | 0.0075 | 0.0050 | 0.0731 | 0.0532 | 0.0183 | 0.0153 | 0.0767 | 0.0600 | 0.0033 | 0.0015 | 0.0475 | 0.0238 |
| + BC Loss | **0.0095*** | **0.0073*** | **0.1194*** | **0.0832*** | **0.0257*** | **0.0227*** | **0.1123*** | **0.0903*** | **0.0077*** | **0.0037*** | **0.0992*** | **0.0510*** |
| Imp. % | 25.0% | 30.1% | 12.1% | 16.9% | 40.4% | 48.4% | 19.3% | 24.7% | 37.5% | 37.0% | 50.3% | 58.4% |

imbalanced or balanced test set is evaluated. However, in real-world applications, the test distributions are usually unavailable and can even reverse the prior in the training distribution. We conjecture that a good debiasing recommender is required to perform well on both imbalanced and balanced test distributions. In our settings, no information about the balanced test is provided in advance.

**Data Splits.** The models are identical across both imbalanced and balanced evaluations. The test distribution in the balanced evaluation is uniform, *i.e.,* randomly sample $15\%$ of interactions with equal probability *w.r.t.* items. Besides, the test splits for the imbalanced test are similarly long-tailed like the train and validation sets, *i.e.,* randomly split the remaining interactions into training, validation, and imbalanced test sets ($60\% : 10\% : 15\%$).

**Results.** Table 1 reports the comparison of performance in imbalanced and balanced test evaluations. The best performing methods are bold and starred, while the strongest baselines are underlined; Imp.% measures the relative improvements of BC loss over the strongest baselines. We observe that:

- **BC loss significantly outperforms the state-of-the-art baselines in both balanced and imbalanced evaluations across all datasets.** In particular, it achieves consistent improvements over the best debiasing baselines and original CF models by $12.1\% \sim 58.4\%$. This clearly demonstrates that BC loss not only effectively alleviates the amplification of popularity bias but also improves the discriminative power of representations. Moreover, Table 5 shows the computational costs of all methods. Compared to the backbone models, BC loss only adds negligible time complexity.

- **Debiasing baselines sacrifice the imbalanced performance and perform inconsistently across datasets.** Debiasing strategies generally achieve higher balanced performance at the expense of a large imbalanced performance drop. Specifically, the strongest baselines over all imbalanced test sets are the original CF models. Worse still, as the degree of data sparsity increases, some debiasing methods fail to quantify the popularity bias and limit their bias removal ability. For example, in the sparsest Alibaba-iFashion dataset, the results of MF+IPS-CN, MF+CausE, LightGCN+MACR, and LightGCN+CausE on the balanced evaluation are lower than original CF models (MF or Light-GCN). In contrast, benefiting from popularity bias-aware margin, BC loss can learn discriminative representations that accomplish more profound user and item understanding, leading to higher head and tail recommendation quality.

### 5.1.2 Evaluations on Temporal Split Test Set

**Motivation.**
In real applications, popularity bias dynamically changes over time. Here we consider temporal split test evaluation on Douban Movie where the historical interactions are sliced into the training, validation, and test sets (7:1:2) according to the timestamps.

Table 2: The performance comparison on Douban dataset.

| | MF | | | LightGCN | | |
| | HR | Recall | NDCG | HR | Recall | NDCG |
|---|---|---|---|---|---|---|
| Backbone | 0.2924 | 0.0294 | 0.0472 | 0.3543 | 0.0313 | 0.0602 |
| + IPS-CN | 0.2514 | 0.0174 | 0.0324 | 0.3212 | 0.0261 | 0.0502 |
| + CausE | 0.2725 | 0.0203 | 0.0376 | 0.3403 | 0.0275 | 0.0514 |
| + sam+reg | 0.2826 | 0.0191 | 0.0390 | 0.2944 | 0.0252 | 0.0488 |
| + MACR | 0.1084 | 0.0087 | 0.0163 | 0.3127 | 0.0271 | 0.0519 |
| + BC loss | **0.3742*** | **0.0324*** | **0.0601*** | **0.3562*** | **0.0346*** | **0.0652*** |
| Imp. % | 28.0% | 10.2% | 27.3% | 0.5% | 10.4% | 8.3% |

**Results.** As Table 2 shows, BC loss is steadily superior to all baselines *w.r.t.* all metrics on Douban Movie. For instance, it achieves

Table 3: The performance evaluations of head, mid, and tail on Tencent dataset.

| | Balanced NDCG@20 | | | | Imbalanced NDCG@20 | | | |
|---|---|---|---|---|---|---|---|---|
| | Tail | Mid | Head | Overall | Tail | Mid | Head | Overall |
| MF | 0.00004 | 0.00097 | 0.01250 | 0.00402 | 0.00021 | 0.00197 | 0.06837 | 0.06431 |
| + IPS-CN | $0.00009^{+125\%}$ | $0.00212^{+119\%}$ | $0.01684^{+35\%}$ | $0.00575^{+43\%}$ | $0.00056^{+167\%}$ | $0.00401^{+104\%}$ | $0.04439^{-35\%}$ | $0.04205^{-35\%}$ |
| + CausE | $0.00008^{+100\%}$ | $0.00149^{+54\%}$ | $0.01168^{-7\%}$ | $0.00430^{+7\%}$ | $0.00038^{+81\%}$ | $0.00253^{+28\%}$ | $0.04876^{-29\%}$ | $0.04680^{-27\%}$ |
| + sam-reg | $0.00006^{+50\%}$ | $0.00135^{+39\%}$ | $0.01573^{+26\%}$ | $0.00535^{+33\%}$ | $0.00011^{-48\%}$ | $0.00281^{+43\%}$ | $0.02850^{-58\%}$ | $0.02661^{-59\%}$ |
| + MACR | $\mathbf{0.00188}^{+4600\%}$ | $\mathbf{0.00521}^{+437\%}$ | $0.00555^{-56\%}$ | $0.00456^{+13\%}$ | $\mathbf{0.00370}^{+1662\%}$ | $0.00615^{+212\%}$ | $0.02748^{-60\%}$ | $0.02413^{-62\%}$ |
| + BC loss | $0.00024^{+500\%}$ | $0.00355^{+266\%}$ | $\mathbf{0.01831}^{+46\%}$ | $\mathbf{0.00680}^{+69\%}$ | $0.00142^{+576\%}$ | $\mathbf{0.00712}^{+261\%}$ | $\mathbf{0.09552}^{+40\%}$ | $\mathbf{0.09040}^{+41\%}$ |
| LightGCN | 0.00025 | 0.00193 | 0.01136 | 0.00417 | 0.00094 | 0.00391 | 0.07561 | 0.07121 |
| + IPS-CN | $0.00140^{+460\%}$ | $0.00241^{+25\%}$ | $0.01560^{+37\%}$ | $0.00544^{+30\%}$ | $0.00109^{+16\%}$ | $0.00522^{+34\%}$ | $0.06333^{-16\%}$ | $0.05993^{-16\%}$ |
| + CausE | $0.00006^{-76\%}$ | $0.00138^{-29\%}$ | $0.01177^{+4\%}$ | $0.00403^{-3\%}$ | $0.00040^{-57\%}$ | $0.00279^{-29\%}$ | $0.06996^{-7\%}$ | $0.06650^{-7\%}$ |
| + sam-reg | $0.00006^{-76\%}$ | $0.00120^{-38\%}$ | $0.01727^{+52\%}$ | $0.00560^{+34\%}$ | $0.00024^{-74\%}$ | $0.00253^{-35\%}$ | $0.04647^{-39\%}$ | $0.04355^{-39\%}$ |
| + MACR | $\mathbf{0.00287}^{+1048\%}$ | $\mathbf{0.00461}^{+139\%}$ | $0.00454^{-60\%}$ | $0.00501^{+20\%}$ | $\mathbf{0.00389}^{+313\%}$ | $\mathbf{0.00635}^{+62\%}$ | $0.04058^{-46\%}$ | $0.05323^{-25\%}$ |
| + BC loss | $0.00057^{+128\%}$ | $0.00321^{+66\%}$ | $\mathbf{0.01943}^{+71\%}$ | $\mathbf{0.00730}^{+75\%}$ | $0.00125^{+33\%}$ | $0.00516^{+32\%}$ | $\mathbf{0.08823}^{+17\%}$ | $\mathbf{0.08320}^{+17\%}$ |

significant improvements over the MF and LightGCN backbones *w.r.t.* Recall@20 by 10.2% and 10.4%, respectively. This validates that BC loss endows the backbone models with better robustness against the popularity distribution shift and alleviates the negative influence of popularity bias. Surprisingly, none of the debiasing baselines could maintain a comparable performance to the backbones. We ascribe the failure to their preconceived idea of tail items, which possibly change over time.

## 5.2 Head, Mid, & Tail Performance (RQ2)

**Motivation.** To further evaluate whether BC loss lifts the tail performance by inevitably sacrificing the head performance, we divide the test set of Tencent into three subgroups, according to the interaction number of each item: head (popular items that are in the top third), mid (normal items in the middle), and tail (unpopular items in the bottom third). Most previous studies focus on average NDCG@20 for evaluation, especially balanced test evaluations [22, 23]. However, average metrics could be insufficient to reflect the performance of each subgroup. A trivial solution to achieve high performance is promoting the rankings of low-popularity items in the recommendations. In this case, only the average metrics are not reliable on the balanced test. Therefore, we report the performance of individual subgroups on both balanced and imbalanced test sets for a more comprehensive comparison.

**Results.** Table 3 shows the evaluations of the head, mid, and tail subgroups. The red and blue numbers in percentage separately refer to the improvement and decline of each method relative to the original CF model (MF or LightGCN). We find that:

- **BC loss is the only method that consistently yields remarkable improvements in every subgroup.** With a closer look at the head evaluation, BC loss shows its ability to learn more discriminative representations for popular items across imbalanced and balanced settings. In particular, it achieves significant improvements over MF and LightGCN *w.r.t.* head NDCG by 40% and 17% in the imbalanced test evaluation, respectively. We attribute improvements to the usage of bias-aware margin, which boosts the recommendation quality for the tail and head items.

- **As the performance comparison among subgraphs in the imbalanced scenario shows, the baselines enhance the tail performance but sacrifice the head performance.** Specifically, these baselines hardly maintain the head performance and show a clear trade-off trend between the head and tail performance. Taking MACR as an example, although the great improvement $(+1662\%)$ over MF is achieved in the tail subgraph, it brings in the dramatic drop $(-62\%)$ in the head subgraph, which lowers the overall performance by a big drop $(-60\%)$. Here we ascribe the trade-off to blindly promoting the rankings of tail items for matched and unmatched, rather than improving the discriminative power of representations.

## 5.3 Study on BC Loss (RQ3)

**Effect of Bias-aware Margin.** Figure 3a displays the performance on balanced and imbalanced test sets on Tencent among softmax loss, BC loss with constant margin M [40], and BC loss with adaptive bias-aware margin. BC loss achieves the best performance, illustrating that bias-aware margin indeed is effective at reducing popularity bias and learning high-quality representations.

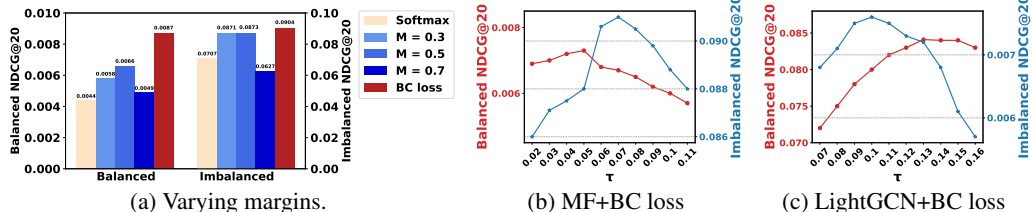

Figure 3: (a) Comparisons with a varying margin; (b-c) Temperature $\tau$ sensitivity analysis on Tencent.

**Effect of Temperature $\tau$.** BC loss has one hyperparameter to tune — temperature $\tau$ in Equation (5). In Figure 3b and 3c, both balanced and imbalanced evaluations exhibit the concave unimodal functions of $\tau$, where the curves reach the peak almost synchronously in a small range of $\tau$. For example, MF+BC loss gets the best performance when $\tau = 0.05$ and $\tau = 0.07$ in balanced and imbalanced settings, respectively; We observe similar trends on other datasets and skip them due to the space limit. This justifies that BC loss does not suffer from the trade-off between the balanced and imbalanced evaluations and improves the generalization without sacrificing the head performance.

## 6 Related Work

Prevalent popularity debiasing strategies in CF roughly fall into four research lines.

**Post-processing re-ranking methods** [5, 6, 7, 8, 9] are applied to the output of the recommender system without changing the representations of users and items. The purposes of modifying the ranking of models can be various: Calibration [5] ensures that the past interests proportions of users are expected to maintain at the same level; RankALS [6] aims to increase the diversification of recommendation; FPC [8] investigates the popularity bias in the dynamic recommendation by rescaling the predicted scores.

**Regularization-based frameworks** [10, 11, 12, 9] explore the use of regularization that provides a tunable mechanism for controlling the trade-off between recommendation accuracy and coverage. The difference among these methods is the design of penalty terms: ALS+Reg [11] defines intra-list distance as the penalty to achieve the fair recommendation; ESAM [10] introduces the attribute correlation alignment, center-clustering, and self-training regularization to learn good feature representations; sam-reg [12] regularizes the biased correlation between user-item relevance and item popularity; Reg [9] decouples the item popularity with the model preference predictions.

**Sample re-weighting methods** [13, 14, 15, 16, 17, 18], also known as Inverse Propensity Score (IPS), view the item popularity in the training set as the propensity score and exploit its inverse to re-weight loss of each instance. To address the high variance of re-weighted loss, many of them [15, 14] further employ normalization or smoothing penalty to attain a more stable output. However, the unreliability of methods is due to their measurement of the propensity score, leveraging the item frequency but failing to consider interaction-wise popularity bias.

**Bias removal by causal inference methods** [19, 24, 23, 20, 21, 22], getting inspiration from the recent success of counterfactual inference, specify the role of popularity bias in assumed causal graphs and mitigate the bias effect on the prediction. However, the causal structure is heuristically assumed based on the author's understanding, without any theoretical guarantee.

BC loss opens up a possibility of conventional debiasing methods in CF that mitigate the popularity bias by enhancing the discriminative power. Recent studies, boosting the discriminative feature spaces by modified softmax loss are mainly discussed in face recognition, where a constant margin is added [40] to better classify. We transfer it in CF and compare it with BC loss in Figure 3a.

## 7 Conclusion

Despite the great success in collaborative filtering, today's popularity debiasing methods are still far from being able to improve the recommendation quality. In this work, we proposed a simple yet effective BC loss, utilizing popularity bias-aware margin to eliminate the popularity bias. Grounded by theoretical proof, clear geometric interpretation and real-world visualization study, BC loss boosts

the head and tail performance by learning a more discriminative representation space. Extensive experiments verify that the remarkable improvement in head and tail evaluations on various test sets indeed comes from the better representation rather than simply catering to the tail.

The limitations of BC loss are in three respects, which will be addressed in future work: 1) the modeling of bias-aware margin is worth exploring, which could significantly influence the performance of BC loss, 2) multiple important biases, such as exposure and selection bias, are not considered, and 3) more experiments comparing BC loss to standard CF losses (*e.g.,* cross-entropy, WARP) are needed to further demonstrate the power of BC loss in regular recommendation tasks (See comparison to BPR, CCL and SSM in Appendix B.4). We believe that this work provides a potential research direction to diagnose the debiasing of long-tail ranking and will inspire more works.

## Acknowledgments and Disclosure of Funding

This research is supported by the Sea-NExT Joint Lab, and CCCD Key Lab of Ministry of Culture and Tourism. Assistance provided by Jingnan Zheng (e0718957@u.nus.edu) is greatly appreciated.

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
