# A  In-depth Analysis of BC loss

## A.1  Visualization of A Toy Experiment

Here we visualize a toy example on the Yelp2018 [25] dataset to showcase the effect of BC loss. Specifically, we train a two-layer LightGCN whose embedding size is three, and illustrate the 3-dimensional normalized representations on a 3D unit sphere in Figure 1 (See the magnified view in Figure 4). We train the identical LightGCN backbone with different loss functions: BPR loss, softmax loss, BC loss, and IPS-CN [15]. For the same head/tail user (*i.e.,* green stars), we plot 500 items in the unit sphere covering all positive items (*i.e.,* red dots) and randomly-selected negative items (*i.e.,* blue dots) from both the training and testing sets. Moreover, the angle distribution (the second row of each subfigure) of positive and negative items for a certain user quantitatively shows the discriminative power of each loss. We observe that:

- **BC loss learns more discriminative representations in both head and tail user cases. Moreover, BC loss learns a more reasonable representation distribution that is locally clustered and globally separated.** As Figures 4d and 4h show, for head and tail users, BC loss encourages around 40% and 55% of positive items to fall into the group closest to user representations, respectively. In other words, these item representations are almost clustered to a small region. BC loss also achieves the smallest distance *w.r.t.* mean positive angle. This verifies that BC loss tends to learn a high similar item/user compactness. Moreover, Figure 4h presents a clear margin between positive and negative items, reflecting a highly-discriminative power. Compared to softmax loss in Figures 4b and 4f, the compactness and dispersion properties of BC loss come from the incorporation of interaction-wise bias-aware margin.

- **The representations learned by standard CF losses - BPR loss and softmax loss - are not discriminative enough.** Under the supervision of BPR and softmax losses, item representations are separably allocated in a wide range of the unit sphere, where blue and red points occupy almost the same space area, as Figures 4a and 4b demonstrate. Furthermore, Figure 4e shows only a negligible overlap between positive and negative items' angle distributions. However, as the negative items are much more than the positive items for the tail user, a small overlap will make many irrelevant items rank higher than the relevant items, thus significantly hindering the recommendation accuracy. Hence, directly optimizing BPR or softmax loss might be suboptimal for the personalized recommendation tasks.

- **IPS-CN, a well-known popularity debiasing method in CF, is prone to lift the tail performance by sacrificing the representation learning for the head.** Compared with BPR loss in Figure 4e, IPS-CN learns better item representations for the tail user, which achieves smaller mean positive angle as illustrated in Figure 4g. However, for the head user in Figure 4c, the positive and negative item representations are mixed and cannot be easily distinguished. Worse still, the representations learned by IPS-CN has a larger mean positive angle for head user compared to BPR loss. This results in a dramatic performance drop for head evaluations.

## A.2  Hard Example Mining Mechanism - one desirable property of BC loss:

We argue that the mechanism of adaptively mining hard interactions is inherent in BC loss, which improves the efficiency and effectiveness of training. Distinct from softmax loss that relies on the predictive scores to mine hard negative samples [56] and leaves the popularity bias untouched, our BC loss considers the interaction-wise biases and adaptively locates hard informative interactions.

Specifically, the popularity bias extractor $\hat{y}_b$ in Equation (3) can be viewed as a hard sample detector. Considering an interaction $(u, i)$ with a high bias degree $cos(\hat{\xi}_{ui})$, we can only use its popularity information to predict user preference and attain a vanishing bias-aware angular margin $M_{ui}$. Hence, interaction $(u, i)$ will plausibly serve as the biased and easy sample, if it involves the active users and popular items. Its close-to-zero margin makes $(u, i)$'s BC loss approach softmax loss, thus downgrading the ranking criteria to match the basic assumption of softmax loss.

In contrast, if the popularity statistics are deficient in recovering user preference via the popularity bias extractor, the interaction $(u, i)$ garners the low bias degree $cos(\hat{\xi}_{ui})$ and exerts the significant margin $M_{ui}$ on its BC loss. Hence, it could work as the hard sample, which typically covers the tail users and items, and yields a more stringent assumption that user $u$ prefers the tail item over the other

popular items by a large margin. Such a significant margin makes the losses more challenging to learn.

In a nutshell, BC loss adaptively prioritizes the interaction samples based on their bias degree and leads the CF model to shift its attention to hard samples, thus improving both head and tail performance, compared with softmax loss (*cf.* Section 5.3).

## A.3 Proof of Theorem 1

**Theorem 1.** *Let $\mathbf{v}_u \doteq \psi(u)$, $\mathbf{v}_i \doteq \phi(i)$, and $\mathbf{c}_u = \frac{1}{|\mathcal{P}_u|} \sum_{i \in \mathcal{P}_u} \mathbf{v}_i$, $\mathbf{c}_i = \frac{1}{|\mathcal{P}_i|} \sum_{u \in \mathcal{P}_i} \mathbf{v}_u$, where $\mathcal{P}_u = \{i | y_{ui} = 1\}$ and $\mathcal{N}_u = \{i | y_{ui} = 0\}$ are the sets of user $u$'s positive and negative items, respectively; $\mathcal{P}_i = \{u | y_{ui} = 1\}$ is the set of item $i$'s positive users. Assuming the representations of users and items are normalized, the minimization of BC loss is equivalent to minimizing a compactness part and a dispersion part simultaneously:*

$$\mathcal{L}_{BC} \geq \underbrace{\sum_{u \in \mathcal{U}} \|\mathbf{v}_u - \mathbf{c}_u\|^2 + \sum_{i \in \mathcal{I}} \|\mathbf{v}_i - \mathbf{c}_i\|^2 - \underbrace{\sum_{u \in \mathcal{U}} \sum_{j \in \mathcal{N}_u} \|\mathbf{v}_u - \mathbf{v}_j\|^2}_{\text{Dispersion part}}}_{\text{Compactness part}} \propto \underbrace{H(\mathbf{V}|Y)}_{\text{Compactness}} \underbrace{-H(\mathbf{V})}_{\text{Dispersion}}. \quad (7)$$

*Proof.* Let the upper-case letter $\mathbf{V} \in \mathbb{V}$ be the random vector of representation and $\mathbb{V} \subseteq \mathbb{R}^d$ be the representation space. We use the normalization assumption of representations to connect cosine and Euclidean distances, *i.e.,* if $\|\mathbf{v}_u\| = 1$ and $\|\mathbf{v}_i\| = 1$, $\mathbf{v}_u^T \mathbf{v}_i = 1 - \frac{1}{2} \|\mathbf{v}_u - \mathbf{v}_i\|^2$, $\forall u, i$.

Let $\mathcal{P}_u = \{i | y_{ui} = 1\}$ be the set of user $u$'s positive items , $\mathcal{P}_i = \{u | y_{ui} = 1\}$ to be the set of item $i$'s positive users, and $\mathcal{N}_u = \{i | y_{ui} = 0\}$ be the set of user $u$'s negative items. Clearly, there exists an upper bound $m$, *s.t.* $-1 < \cos(\hat{\theta}_{ui} + M_{ui}) \leq \mathbf{v}_u^T \mathbf{v}_i - m < 1$. Therefore, we can analyze BC loss, which has the following relationships:

$$\mathcal{L}_{BC} \geq - \sum_{(u,i) \in \mathcal{O}^+} \log \frac{\exp\left((\mathbf{v}_u^T \mathbf{v}_i - m)/\tau\right)}{\exp\left((\mathbf{v}_u^T \mathbf{v}_i - m)/\tau\right) + \sum_{j \in \mathcal{N}_u} \exp\left((\mathbf{v}_u^T \mathbf{v}_j)/\tau\right)}$$

$$= - \sum_{(u,i) \in \mathcal{O}^+} \frac{\mathbf{v}_u^T \mathbf{v}_i - m}{\tau} + \sum_{(u,i) \in \mathcal{O}^+} \log(\exp \frac{\mathbf{v}_u^T \mathbf{v}_i - m}{\tau} + \sum_{j \in \mathcal{N}_u} \exp \frac{\mathbf{v}_u^T \mathbf{v}_j}{\tau}). \quad (8)$$

We now probe into the first term in Equation (8):

$$- \sum_{(u,i) \in \mathcal{O}^+} \frac{\mathbf{v}_u^T \mathbf{v}_i - m}{\tau} = \sum_{(u,i) \in \mathcal{O}^+} \frac{\|\mathbf{v}_u - \mathbf{v}_i\|^2}{2\tau} + \frac{m-1}{\tau}$$

$$\overset{c}{=} \sum_{(u,i) \in \mathcal{O}^+} \|\mathbf{v}_u - \mathbf{v}_i\|^2$$

$$= \sum_{u \in \mathcal{U}} \sum_{i \in \mathcal{P}_u} (\|\mathbf{v}_u\|^2 - \mathbf{v}_u^T \mathbf{v}_i) + \sum_{i \in \mathcal{I}} \sum_{u \in \mathcal{P}_i} (\|\mathbf{v}_i\|^2 - \mathbf{v}_u^T \mathbf{v}_i)$$

$$\overset{c}{=} \sum_{u \in \mathcal{U}} \|\mathbf{v}_u - \mathbf{c}_u\|^2 + \sum_{i \in \mathcal{I}} \|\mathbf{v}_i - \mathbf{c}_i\|^2, \quad (9)$$

where the symbol $\overset{c}{=}$ indicates equality up to a multiplicative and/or additive constant; $\mathbf{c}_u = \frac{1}{|\mathcal{P}_u|} \sum_{i \in \mathcal{P}_u} \mathbf{v}_i$ is the averaged representation of all items that $u$ has interacted with, which describes $u$'s interest; $\mathbf{c}_i = \frac{1}{|\mathcal{P}_i|} \sum_{u \in \mathcal{P}_i} \mathbf{v}_u$ is the averaged representation of all users who have adopted item $i$, which profiles its user group. We further analyze Equation (9) from the entropy view by conflating the first two terms:

$$- \sum_{(u,i) \in \mathcal{O}^+} \frac{\mathbf{v}_u^T \mathbf{v}_i - m}{\tau} \overset{c}{=} \sum_{\mathbf{v}} \|\mathbf{v} - \mathbf{c}_v\|^2, \quad (10)$$

where $\mathbf{v} \in \{\mathbf{v}_u | u \in \mathcal{U}\} \cup \{\mathbf{v}_i | i \in \mathcal{I}\}$ summarizes the representations of users and items, with the mean of $\mathbf{c}_v$. Following [57], we further interpret this term as a conditional cross-entropy between

$\mathbf{V}$ and another random variable $\bar{\mathbf{V}}$ whose conditional distribution given $Y$ is a standard Gaussian $\bar{\mathbf{V}}|Y \sim \mathcal{N}(\mathbf{c_V}, I)$:

$$- \sum_{(u,i)\in\mathcal{O}^+} \frac{\mathbf{v}_u^T \mathbf{v}_i - m}{\tau} \stackrel{\text{c}}{=} H(\mathbf{V}; \bar{\mathbf{V}}|Y) = H(\mathbf{V}|Y) + D_{KL}(\mathbf{V}||\bar{\mathbf{V}}|Y)$$

$$\propto H(\mathbf{V}|Y), \tag{11}$$

where $H(\cdot)$ denotes the cross-entropy, and $D_{KL}(\cdot)$ denotes the $KL$-divergence. As a consequence, the first term in Equation (8) is positive proportional to $H(\mathbf{V}|Y)$. This concludes the proof for the first compactness part of BC loss.

We then inspect the second term in Equation (8) to demonstrate its dispersion property:

$$\sum_{(u,i)\in\mathcal{O}^+} \log(\exp\frac{\mathbf{v}_u^T\mathbf{v}_i - m}{\tau} + \sum_{j\in\mathcal{N}_u} \exp\frac{\mathbf{v}_u^T\mathbf{v}_j}{\tau})$$

$$\geq \sum_{(u,i)\in\mathcal{O}^+} \log\big(\sum_{j\in\mathcal{N}_u} \exp\frac{\mathbf{v}_u^T\mathbf{v}_j}{\tau}\big)$$

$$\geq \sum_{u\in\mathcal{U}} \sum_{j\in\mathcal{N}_u} \frac{\mathbf{v}_u^T\mathbf{v}_j}{\tau}$$

$$\stackrel{\text{c}}{=} - \sum_{u\in\mathcal{U}} \sum_{j\in\mathcal{N}_u} \|\mathbf{v}_u - \mathbf{v}_j\|^2, \tag{12}$$

where we drop the redundant terms aligned with the compactness objective in the second line, and adopt Jensen's inequality in the third line. As shown in prior studies [58], minimizing this term is equivalent to maximizing entropy $H(\mathbf{V})$:

$$\sum_{(u,i)\in\mathcal{O}^+} \log(\exp\frac{\mathbf{v}_u^T\mathbf{v}_i - m}{\tau} + \sum_{j\in\mathcal{N}_u} \exp\frac{\mathbf{v}_u^T\mathbf{v}_j}{\tau}) \propto -\mathcal{H}(\mathbf{V}). \tag{13}$$

As a result, the second term in Equation (8) works as the dispersion part in BC loss. $\square$

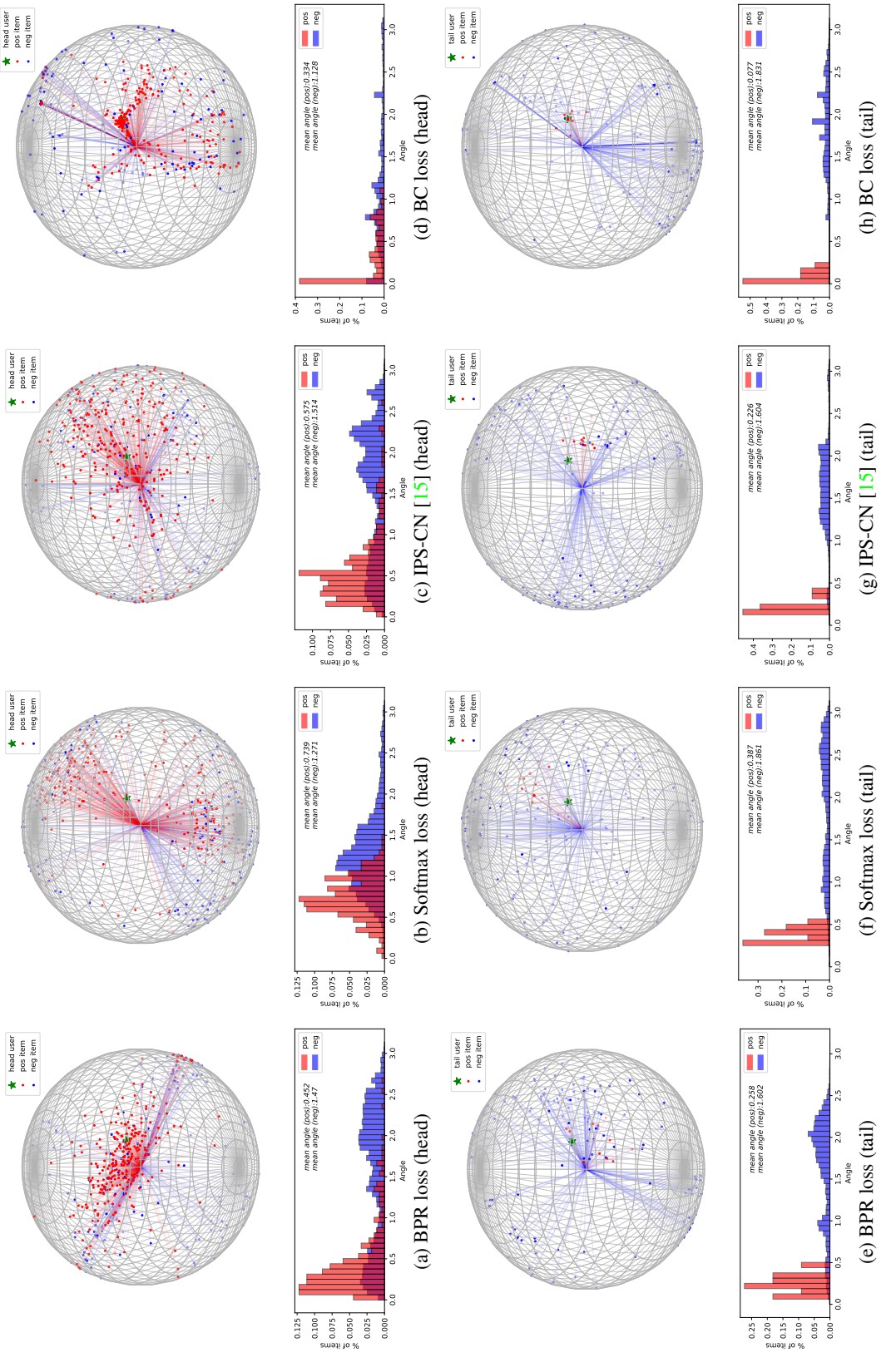

Figure 4: Magnified View of Figure 1.

# B  Experiments

## B.1  Experimental Settings

**Datasets.** We conduct experiments on eight real-world benchmark datasets: Tencent [47], Amazon-Book [48], Alibaba-iFashion [49], Yelp2018 [25], Douban Movie [50], Yahoo!R3 [51], Coat [13] and KuaiRec [52]. All datasets are public and vary in terms of size, domain, and sparsity. Table 4 summarizes the dataset statistics, where the long-tail degree is monitored by KL-divergence between the item popularity distribution and the uniform distribution, *i.e.,* $D_{KL}(\hat{P}_{data}||\text{Uniform})$. A larger KL-divergence value indicates that the heavier portion of interactions concentrates on the head of distribution. In the stage of pre-processing data, we follow the standard 10-core setting [20, 59] to filter out the items and users with less than ten interactions.

**Data Splits.** For comprehensive comparisons, almost all standard test distributions in CF are covered in the experiments: balanced test set [22, 23, 24], randomly selected imbalanced test set [10, 53], temporal split test set [20, 21, 12], and unbiased test set [13, 52, 51]. Three datasets (*i.e.,* Tencent, Amazon-Book, Alibaba-iFashion) are partitioned into both balanced and randomly selected imbalanced evaluations. As an intervention test, the balanced evaluation (*i.e.,* uniform distribution) is frequently employed in recent debiasing CF approaches [22, 23, 24, 60]. Douban is split based on the temporal splitting strategy [61]. KuaiRec is an unbiased, fully-observed dataset in which the feedback of the test set's interaction is explicitly collected.

**Evaluation Metrics.** We adopt the all-ranking strategy [62], *i.e.,* for each user, all items are ranked by the recommender model, except the positive ones in the training set. To evaluate the quality of recommendation, three widely-used metrics are used: Hit Ratio (HR@$K$), Recall@$K$, Normalized Discounted Cumulative Gain (NDCG@$K$), where $K$ is set as 20 by default.

**Baselines.** We validate our BC loss on two widely-used CF models, MF [26] and LightGCN [25], which are representatives of the conventional and state-of-the-art CF models. We compare BC loss with the popular debiasing strategies in various research lines: sample re-weighting (IPS-CN [15]), bias removal by causal inference (MACR [22], CausE [19]), and regularization-based framework (sam+reg [12]). We also compare BC loss to other standard losses used in collaborative filtering including most commonly used loss (BPR loss [26]) and newest proposed softmax losses (CCL [54] and SSM [55]).

**Parameter Settings.** We conduct experiments using a Nivida-V100 GPU (32 GB memory) on a server with a 40-core Intel CPU (Intel(R) Xeon(R) CPU E5-2698 v4). We implement our BC loss in PyTorch. Our codes, datasets, and hyperparameter settings are available at https://github.com/anzhang314/BC-Loss to guarantee reproducibility. For a fair comparison, all methods are optimized by Adam [63] optimizer with the batch size as 2048, embedding size as 64, learning rate as 1e-3, and the coefficient of regularization as 1e-5 in all experiments. Following the default setting in [25], the number of embedding layers for LightGCN is set to 2. We adopt the early stop strategy that stops training if Recall@20 on the validation set does not increase for 10 successive epochs. A grid search is conducted to tune the critical hyperparameters of each strategy to choose the best models *w.r.t.* Recall@20 on the validation set. For softmax, SSM, and BC loss, we search $\tau$ in [0.06, 0.14] with a step size of 0.02. For CausE, 10% of training data with balanced distribution are used as intervened set, and $cf\_pen$ is tuned in $[0.01, 0.1]$ with a step size of 0.02. For MACR, we follow the

Table 4: Dataset statistics.

|  | KuaiRec | Douban Movie | Tencent | Amazon-Book | Alibaba-iFashion | Yahoo!R3 | Coat |
|---|---|---|---|---|---|---|---|
| #Users | 7175 | 36,644 | 95,709 | 52,643 | 300,000 | 14382 | 290 |
| #Items | 10611 | 22,226 | 41,602 | 91,599 | 81,614 | 1000 | 295 |
| #Interactions | 1062969 | 5,397,926 | 2,937,228 | 2,984,108 | 1,607,813 | 129,748 | 2,776 |
| Sparsity | 0.01396 | 0.00663 | 0.00074 | 0.00062 | 0.00007 | 0.00902 | 0.03245 |
| $D_{KL}$-Train | 1.075 | 1.471 | 1.425 | 0.572 | 1.678 | 0.854 | 0.356 |
| $D_{KL}$-Validation | 1.006 | 1.642 | 1.423 | 0.572 | 1.705 | 0.822 | 0.350 |
| $D_{KL}$-Balanced | - | - | 0.003 | 0.000 | 0.323 | - | - |
| $D_{KL}$-Imbalanced | - | - | 1.424 | 0.571 | 1.703 | - | - |
| $D_{KL}$-Temporal | - | 1.428 | - | - | - | - | - |
| $D_{KL}$-Unbiased | 1.666 | - | - | - | - | 0.100 | 0.109 |

Table 5: Training cost on Tencent (seconds per epoch/in total).

|  | Backbone | +IPS-CN | +CausE | +sam+reg | +MACR | +BC loss |
|---|---|---|---|---|---|---|
| MF | 15.5 / 17887 | 17.8 / 10662 | 16.6 / 1859 | 18.2 / 3458 | 160 / 17600 | 36.1 / 12815 |
| LightGCN | 78.6 / 4147 | 108 / 23652 | 47.2 / 3376 | 49.8 / 10458 | 135 / 20250 | 283 / 7075 |

Table 6: Performance comparison on KuaiRec dataset.

|  | Validation | | | Unbiased Test | | |
|---|---|---|---|---|---|---|
|  | HR | Recall | NDCG | HR | Recall | NDCG |
| LightGCN | 0.299 | 0.069 | 0.051 | 0.104 | 0.0038 | 0.0064 |
| + IPS-CN | 0.255 | 0.056 | 0.042 | 0.109 | 0.0073 | 0.0083 |
| + CausE | 0.292 | 0.067 | 0.050 | 0.101 | 0.0056 | 0.0077 |
| + sam+reg | 0.274 | 0.060 | 0.047 | 0.107 | 0.0069 | 0.0080 |
| + BC loss | 0.343 | 0.076 | 0.062 | **0.139*** | **0.0077*** | **0.0115*** |
| Imp.% | - | - | - | 27.5% | 4.05% | 38.6% |

original settings to set weights for user branch $\alpha = 1e - 3$ and item branch $\beta = 1e - 3$, respectively. We further tune hyperparameter $c = [0, 50]$ with a step size of 5. For CCL loss, we search $w$ in $\{1, 2, 5, 10, 50, 100, 200\}$, $m$ in the range $[0.2, 1]$ with a step size of 0.2. When it come to the number of negative samples, the softmax, SSM, CCL, and BC loss set 128 for MF backbone and in-batch negative sampling for LightGCN models.

## B.2 Training Cost

In terms of time complexity, as shown in Table 5, we report the time cost per epoch and in total of each baselines on Tencent. Compared with backbone methods (*i.e.,* MF and LightGCN), BC loss adds very little computing complexity to the training process.

## B.3 Evaluations on Unbiased Test Set

**Motivation.** Because of the missing-not-at-random condition in a real recommender system, offline evaluation on collaborative filtering and recommender system is commonly acknowledged as a challenge. To close the gap, Yahoo!R3 [51] and Coat [13] are widely used, which offer unbiased test sets that are collected using the missing-complete-at-random (MCAR) concept. Additionally, newly proposed KuaiRec [52] also provides a fully-observed unbiased test set with 1,411 users over 3,327 videos. We conduct experiments on all these datasets for comprehensive comparison, and KuaiRec is also included as one of our unbiased evaluations for two key reasons: 1) It is significantly larger than existing MCAR datasets (*e.g.,* Yahoo! and Coat); 2) It overcomes the missing values problem, making it as effective as an online A/B test.

**Parameter Settings.** For BC loss on Yahoo!R3 and Coat, we search $\tau_1$ in [0.05, 0.21] with a step size of 0.01, and $\tau_2$ in [0.1, 0.6] with a step size of 0.1, and search the number of negative samples in [16, 32, 64, 128]. We adopt the batch size of 1024 and learning rate of 5e-4.

**Results.** Table 6 and 8 illustrate the unbiased evaluations on KuaiRec, Yahoo!R3, and Coat dataset using LightGCN and MF as backbone models, respectively. The best performing methods are bold and starred, while the strongest baselines are underlined; Imp.% measures the relative improvements of BC loss over the strongest baselines. BC loss is consistently superior to all baselines *w.r.t.* all metrics. It indicates that BC loss truly improves the generalization ability of recommender.

## B.4 Performance Comparison with Standard Loss Functions in CF

**Motivation.** To verify the effectiveness of BC loss as a standard learning strategy in collaborative filtering, we further conduct the experiments over various datasets between BC loss, BPR loss, CCL loss [54], and SSM loss [55]. We choose these three losses as baselines for two main reasons: 1) BPR loss is the most commonly applied in recommender system; 2) SSM and CCL are most recent proposed losses, where SSM is also a softmax loss and CCL employs a global margin.

Table 7: Performance comparison on Tecent, Amazon-book and Alibaba-iFashion datasets.

| | Tencent | | | | Amazon-book | | | | Alibaba-iFashion | | | |
| | Balanced | | Imbalanced | | Balanced | | Imbalanced | | Balanced | | Imbalanced | |
| | Recall | NDCG | Recall | NDCG | Recall | NDCG | Recall | NDCG | Recall | NDCG | Recall | NDCG |
|---|---|---|---|---|---|---|---|---|---|---|---|---|
| BPR | 0.0052 | 0.0040 | 0.0982 | 0.0643 | 0.0109 | 0.0103 | 0.0850 | 0.0638 | 0.0056 | 0.0028 | 0.0843 | 0.0411 |
| SSM | 0.0055 | 0.0045 | 0.1297 | 0.0872 | 0.0156 | 0.0157 | 0.1125 | 0.0873 | 0.0079 | 0.0040 | 0.0963 | 0.0436 |
| CCL | 0.0057 | 0.0047 | 0.1216 | 0.0818 | 0.0175 | 0.0167 | 0.1162 | 0.0927 | 0.0075 | 0.0038 | 0.0954 | 0.0428 |
| BC Loss | 0.0087* | 0.0068* | 0.1298* | 0.0904* | 0.0221* | 0.0202* | 0.1198* | 0.0948* | 0.0095* | 0.0048* | 0.0967* | 0.0487* |
| Imp. % | 52.6% | 44.7% | 0.1% | 3.7% | 26.3% | 21.0% | 3.1% | 2.3% | 20.3% | 20.0% | 0.4% | 11.7% |

Table 8: Performance comparison on Yahoo!R3 and Coat dataset.

| | Yahoo!R3 | | Coat | |
| | Recall | NDCG | Recall | NDCG |
|---|---|---|---|---|
| IPS-CN | 0.1081 | 0.0487 | 0.1700 | 0.1377 |
| CausE | 0.1252 | 0.0537 | 0.2329 | 0.1635 |
| sam+reg | 0.1198 | 0.0548 | 0.2303 | 0.1869 |
| MACR | 0.1243 | 0.0539 | 0.0798 | 0.0358 |
| BPR | 0.1063 | 0.0476 | 0.0741 | 0.0361 |
| SSM | 0.1470 | 0.0688 | 0.2022 | 0.1832 |
| CCL | 0.1428 | 0.0676 | 0.2150 | 0.1885 |
| BC loss | 0.1487* | 0.0706* | 0.2385* | 0.1969* |
| Imp.% | 1.2% | 2.6% | 2.4% | 4.5% |

**Results.** Table 7 reports the performance on both balanced and imbalanced test sets on various datasets among different losses. We have two main observations: (1) Clearly, our BC loss consistently outperforms CCL and SSM; (2) CCL and SSM achieve comparable performance to BC loss in the imbalanced evaluation settings, while performing much worse than BC loss in the balanced evaluation settings. This indicates the superiority of BC loss in alleviating the popularity bias, and further justifies the effectiveness of the bias-aware margins.

Table 8 shows the unbiased evaluations on Yahoo!R3 and Coat dataset. With regard to all criteria, BC loss constantly outperforms all other losses. It verifies the effectiveness of instance-wise bias margin of BC loss.

## B.5 Study on BC Loss (RQ3) - Effect of Popularity Bias Extractor

**Motivation.** To check the effectiveness of popularity bias extractor, we need to answer two main questions: 1) what kinds of interactions will be learned well by the bias extractor? What does the learned bias-angle distribution look like? 2) can BC loss benefit from the bias margin extracted according to the popularity bias extractor in various groups of interactions? We devise the following experiment to tackle the aforementioned problems.

**Experiment Setting.** Users can be divided into three parts: head, mid, and tail, based on their popularity scores. Analogously, items can be partitioned into the head, mid, and tail parts. As such, we can categorize all user-item interactions into nine subgroups. We have visualized the learned angles for various types of interactions in the following table 5.

**Results.** The table 5 shows the learned angles over all subgroups. We find that interactions between head users and head items tend to hold small angles. Moreover, as evidenced by the high standard deviation, the interactions stemming from the same subgroup types are prone to receive a wide range of angular values. This demonstrates the variability and validity of instance-wise angular margins.

## B.6 Visualization of interaction bias degree

Figures 6a and 6b illustrate the relations between popularity scores and bias degree extracted by popularity bias extractor *w.r.t.* user and item sides, respectively. Specifically, positive trends are shown, where their relations are also quantitatively supported by Pearson correlation coefficients. (0.7703 and 0.662 for item and user sides, respectively). It verifies the power of popularity embeddings to predict the popularity scores — that is, user popularity embeddings derived from the popularity bias extractor are strongly correlated and sufficiently predictive to user popularity scores; analogously to the item side.

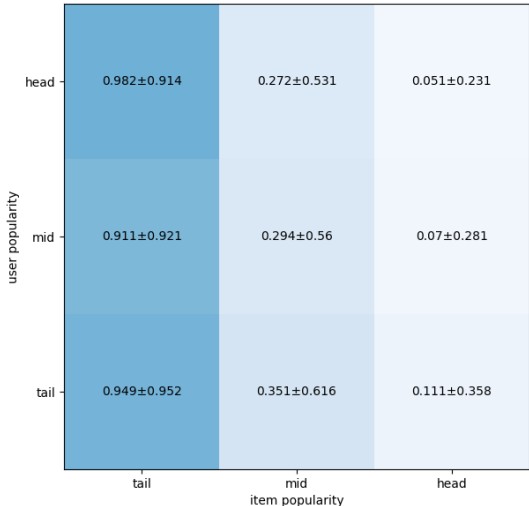

Figure 5: Visualization of popularity bias angle for different types of interactions on Tencent.

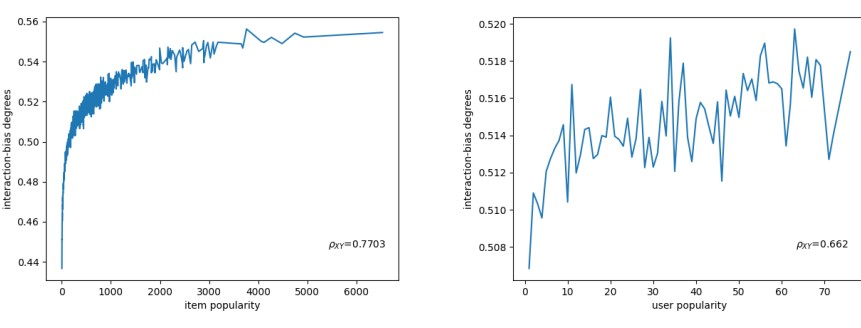

(a) Bias degree *w.r.t.* Item popularity bias

(b) Bias degree *w.r.t.* User popularity bias

Figure 6: Visualizations of relationships between interaction bias degree estimated by our popularity bias extractor and item/user popularity statistics. Pearson correlation coefficients are provided.

Table 9: Model architectures and hyper-parameters

|  | | BC loss hyper-parameters | | | |
|---|---|---|---|---|---|
|  | $\tau_1$ | $\tau_2$ | lr | batch size | No. negative samples |
| **MF** | | | | | |
| Tencent | 0.06 | 0.1 | 1e-3 | 2048 | 128 |
| iFashion | 0.08 | 0.1 | 1e-3 | 2048 | 128 |
| Amazon | 0.08 | 0.1 | 1e-3 | 2048 | 128 |
| Douban | 0.08 | 0.1 | 1e-3 | 2048 | 128 |
| Yahoo!R3 | 0.15 | 0.2 | 5e-4 | 1024 | 128 |
| Coat | 0.09 | 0.4 | 5e-4 | 1024 | 64 |
| **LightGCN** | | | | | |
| Tencent | 0.12 | 0.1 | 1e-3 | 2048 | in-batch |
| iFashion | 0.14 | 0.1 | 1e-3 | 2048 | in-batch |
| Amazon | 0.08 | 0.1 | 1e-3 | 2048 | in-batch |
| Douban | 0.14 | 0.1 | 1e-3 | 2048 | in-batch |