# OpenReview forum: "Incorporating Bias-aware Margins into Contrastive Loss for Collaborative Filtering"
_NeurIPS.cc/2022/Conference — NeurIPS 2022 Accept_

### Official Review · Reviewer_aXKZ · 2022-07-11

**Rating:** 8
**Confidence:** 4
**Soundness:** 3 good
**Presentation:** 4 excellent
**Contribution:** 3 good

**Summary:**

This work tries to reduce the negative impact of popularity bias on CF models. The paper incorporates Bias-aware margins into Contrastive loss and proposes a simple yet effective BC Loss, where the margin tailors quantitatively to the bias degree of each user-item interaction. Extensive experiments on six datasets demonstrate the effectiveness of the proposed algorithm.

**Questions:**

Please refer to my questions above.

**Ethics Review Area:**

["I don’t know"]

**Limitations:**

The limitations are discussed in the Section 7. I’m not aware of any potential negative societal impact.



**Strengths And Weaknesses:**

For the strengths, following the idea of MACR, this paper also designs a bias extractor and aims to remove the popularity bias in practice. Interestingly, this work explored the debias problem from a geometric view and unified the bias term into a marginal contrastive learning framework elegantly. The geometric interpretation and theoretical analysis also show the good properties of the proposed BC loss. Extensive experiments on six datasets with various settings verify the effectiveness of BC loss on both debiasing and general recommendation. Overall, I agree this is a good paper with clear novelty.

Meanwhile, I also have some questions. For the Section Popularity Bias Extractor, I wonder if it'll lose part of useful bias information if just inputting the user/item popularity score. Have you tried other inputs? It's also unclear what kinds of interactions will be learned well according to equation (4). It'll be better if you can provide some visualizations of the learned angles of different types of interactions in the experiments. For the experiments, I'd like to know if the balanced test setting uses an imbalanced or balanced validation set. Though current experiments demonstrate the ability of BC loss on debiasing. however, to verify its effectiveness as a standard loss, more comparisons with some general loss functions are required.

---

> ### Author Response · Authors · 2022-08-02
> **Response to Reviewer aXKZ - Part 1**
>
> Thanks so much for your time and positive feedback! To address your concerns, we present the point-to-point responses as follows.
>
> > **Comment 1: Extend the Popularity Bias Extractor** - “For the Section Popularity Bias Extractor, I wonder if it'll lose part of useful bias information if just inputting the user/item popularity score. Have you tried other inputs?”
>
> Thanks for your constructive comment. Extending the popularity bias extractor with different inputs is an interesting and promising research direction. We have fed the popularity statistics of a user and an item into a MLP to get the bias degree. However, the debiasing performance drops dramatically. This indicates that inputting the popularity scores of the user and item hardly represents the interaction-wise bias. We leave the exploration of bias extractors in future work.
>
> > **Comment 2: Visualization of Popularity Bias Angle** - “It's also unclear what kinds of interactions will be learned well according to equation (4). It'll be better if you can provide some visualizations of the learned angles of different types of interactions in the experiments.”
>
> We appreciate your insightful suggestions. We have visualized the learned angles for various types of interactions in the following table. Users can be divided into three parts: head, mid, and tail, based on their popularity scores. Analogously, items can be partitioned into the head, mid, and tail parts. As such, we can categorize all user-item interactions into nine subgroups. The following table shows the learned angles over all subgroups. We find that interactions between head users and head items tend to hold small angles. Moreover, as evidenced by the high standard deviation, the interactions stemming from the same subgroup types are prone to receive a wide range of angular values. This demonstrates the variability and validity of instance-wise angular margins.
>
> |               | **Head User** | **Mid User**  | **Tail User** |
> |---------------|---------------|---------------|---------------|
> | **Head Item** | 0.051 ± 0.231 | 0.07 ± 0.281  | 0.111 ± 0.358 |
> | **Mid Item**  | 0.272 ± 0.531 | 0.294 ± 0.560 | 0.351 ± 0.616 |
> | **Tail Item** | 0.982 ± 0.914 | 0.911 ± 0.921 | 0.949 ± 0.952 |
>
>
> > **Comment 3** - “For the experiments, I'd like to know if the balanced test setting uses an imbalanced or balanced validation set.”
>
> Thanks. In our experiments, the balanced test setting uses an imbalanced validation set. We would like to emphasize that no distribution information about the balanced test is provided in advance.
>
>
>
> > **Comment 4: More experiments with other Losses** - “Though current experiments demonstrate the ability of BC loss on debiasing. however, to verify its effectiveness as a standard loss, more comparisons with some general loss functions are required.”
>
> Thanks so much for your suggestions. To address your concerns, we have selected two recent works, CCL [1] and SSM [2], as the general and advanced loss functions. We have conducted additional experiments on Tencent, Amazon, iFashion, Yahoo!R3, and Coat datasets. The performance comparison between BPR, CCL, SSM, and BC loss is summarized in the following tables.
>
> **Table 7 in Appendix: The performance comparison of loss functions on Tencent, Amazon, and iFashion**
> |             |   Tencent  |            |            |            |   Amazon   |            |            |            |  iFashion  |            |            |            |
> |-------------|:----------:|:----------:|:----------:|:----------:|:----------:|:----------:|:----------:|:----------:|:----------:|:----------:|:----------:|:----------:|
> |             |  Balanced  |            | Imbalanced |            |  Balanced  |            | Imbalanced |            |  Balanced  |            | Imbalanced |            |
> |             |   Recall   |    NDCG    |   Recall   |    NDCG    |   Recall   |    NDCG    |   Recall   |    NDCG    |   Recall   |    NDCG    |   Recall   |    NDCG    |
> | BPR         |   0.0052   |   0.0040   |   0.0982   |   0.0643   |   0.0109   |   0.0103   |   0.0850   |   0.0638   |   0.0056   |   0.0028   |   0.0843   |   0.0411   |
> | SSM         |   0.0055   |   0.0045   |   0.1297   |   0.0872   |   0.0156   |   0.0157   |   0.1125   |   0.0873   |   0.0079   |   0.0040   |   0.0963   |   0.0436   |
> | CCL         |   0.0057   |    00047   |   0.1216   |   0.0818   |   0.0175   |   0.0167   |   0.1162   |   0.0927   |   0.0075   |   0.0038   |   0.0954   |   0.0428   |
> | BC Loss | **0.0087** | **0.0068** | **0.1298** | **0.0904** | **0.0221** | **0.0202** | **0.1198** | **0.0948** | **0.0095** | **0.0048** | **0.0967** | **0.0487** |
> | Imp. %      |    52.6%   |    44.7%   |    0.1%    |    3.7%    |    26.3%   |    21.0%   |    3.1%    |    2.3%    |    20.3%   |    20.0%   |    0.4%    |    11.7%   |

---

> > ### Author Response · Authors · 2022-08-02
> > **Response to Reviewer aXKZ - Part 2**
> >
> > **Table 8 in Appendix: Performance comparison on Yahoo!R3 and Coat**
> > |         |  Yahoo!R3  |            |    Coat    |            |
> > |---------|:----------:|:----------:|:----------:|:----------:|
> > |         |   Recall   |    NDCG    |   Recall   |    NDCG    |
> > | BPR     |   0.1063   |   0.0476   |   0.0741   |   0.0361   |
> > | SSM     |   0.1470   |   0.0688   |   0.2022   |   0.1832   |
> > | CCL     |   0.1428   |   0.0676   |   0.2150   |   0.1885   |
> > | BC Loss | **0.1487** | **0.0706** | **0.2385** | **0.1969** |
> > | Imp. %  |    1.2%    |    2.6%    |    2.4%    |    4.5%    |
> >
> > [1] SimpleX: A Simple and Strong Baseline for Collaborative Filtering. 2021.
> > [2] On the Effectiveness of Sampled Softmax Loss for Item Recommendation. 2022

---

> > ### Comment · Reviewer_aXKZ · 2022-08-08
> > **Response to Authors**
> >
> > Thanks very much for your response. I appreciate the time and efforts you put on rebuttal. The experimental compasions with more general losses are pretty impressive, but I have a question that why did models can also achieve substantial improvements on the ImBalanced data set. From my understanding, a model that focuses more on tail items will get worse performance on the Imbalanced Data. The table on popularity bias angles is not very informative but I understand it's hard to visualize and explain the model well. Thanks for your hard work. I still hold positive feedback for this paper.

---

> > > ### Author Response · Authors · 2022-08-09
> > > **Discussion - Why BC loss also achieve improvement on the imbalanced test**
> > >
> > > We appreciate you posing this crucial and fundamental question in the popularity debiasing task. We are excited to discuss this question with you and share our thoughts, since this point, in our mind, is the most insightful takeaway from our work.
> > >
> > > It seems counterintuitive to claim that the popularity debiasing method (also known as the long-tail technique) can perform well on both imbalanced and balanced distributions, although **some approaches in other research domains have yielded similar results** [1,2]. Hence, answering “Why our BC loss can perform well on different distributions w.r.t. popularity?” is the key to revealing our key insights --- **distinct from the trade-off gaming between different distributions performed by prior studies, our BC loss enhances the discriminative power of representations, regardless of popularity shift.**
> > >
> > > Specifically, the trade-off gaming performed by prior studies is to raise the importance of tails to perform better on the balanced distribution, while sacrificing head performance:
> > > - Post-processing re-ranking methods simply lift the tail performance, without touching the representations.
> > > - Most methods designed for balanced distribution will inevitably degrade head performance. (1) For IPS families, they implicitly assume a balanced item distribution w.r.t. users. As a result, IPS tends to perform better when the testing distribution is close to the uniform distribution. (2) For bias removal by causal inference, MACR is one of the SOTA works. Its hyperparameter $c$ is a measure of distribution to eliminate the effect of popularity bias. However, to fine-tune $c$, it needs a balanced validation dataset that is consistent with the test dataset.
> > > - Another major research line is regularization-based methods, such as sam-reg, which is a trade-off with a lambda in the middle to adjust the relative proportion of importance.
> > >
> > > **Our BC loss couples the generalization ability of model with the popularity debiasing idea.** It makes the backbone learn more discriminative representations, regardless of popularity distribution shifts. Figure 1 in our paper visualizes the real three-dimension representations learned by LightGCN on Yelp, which supports our claim that BC loss learns a more discriminative representation space that is locally clustered and globally separated. Experiments in Section 5.2 verify the performance improvement of BC loss in head, mid, and tail subgroups. Moreover, the unbiased test evaluations on KuaiRec, Yahoo!R3, and Cost verify the effectiveness of BC loss in OOD scenarios. In a nutshell, extensive experiments indicate that BC loss can yield good performance w.r.t. OOD distributions.
> > >
> > > Here we would like to emphasize two points:
> > > - Compared to conventional debiasing techniques, BC loss does achieve substantial improvements in imbalanced evaluations. However, compared to some strong variants of softmax loss, such as SSM and CCL, BC loss can maintain competitive performance in imbalanced tests, while significantly superior in balanced tests.
> > > - When broadening our view to OOD research in machine learning, **the conclusion is similar to our results** --- that is, **well-designed OOD methods can generalize better in both in-distribution (imbalanced) and out-of-distributions (balanced)**. For example, recent methods [3,4,5] have shown better results in the worst group (i.e. OOD), while being able to approach or maintain the performance of the SOTA method on average accuracy.
> > >
> > > Moreover, in recommendation, one WWW’2022 paper [6] also supports our view. A model fundamentally has stronger representation learning power, leading to better generalization capabilities. In the pursuit of debiasing techniques that excel on balanced datasets, in our opinion, is not our ultimate goal. **Taking popularity as an inductive bias to learn more generalizable models may be a potential solution to the popularity debiasing problem.**
> > >
> > > We're not sure if you'll agree with us. Although it appears that our conversation has gone far beyond our paper itself, we would be more than delighted to discuss it with you further if you hold a different point of view. Thanks again.

---

> > > > ### Author Response · Authors · 2022-08-09
> > > > **Reference**
> > > >
> > > > [1] Introspective Distillation for Robust Question Answering. NeurIPS 2021
> > > >
> > > > [2] Cross-Domain Empirical Risk Minimization for Unbiased Long-tailed Classification. AAAI 2021.
> > > >
> > > > [3] Improving Out-of-Distribution Robustness via Selective Augmentation. ICML 2022.
> > > >
> > > > [4] DORO: Distributional and Outlier Robust Optimization. ICML 2022.
> > > >
> > > > [5] Distributionally Robust Neural Networks for Group Shifts: On the Importance of Regularization for Worst-Case Generalization. ICLR 2020.
> > > >
> > > > [6] Distributionally-robust Recommendations for Improving Worst-case User Experience. WWW 2022.

---

### Official Review · Reviewer_jXGF · 2022-07-12

**Rating:** 6
**Confidence:** 4
**Soundness:** 3 good
**Presentation:** 4 excellent
**Contribution:** 3 good

**Summary:**

This paper proposes a simple-yet-effective loss, called Bias-aware margins into Contrastive (BC) loss, to mitigate the popularity bias. Specifically, it introduces a popularity bias extract to estimate the popularity bias scores from a pair of user and item popularity. Then it incorporates it into the contrastive loss as the marginal information. Despite the simplicity of the proposed loss, experimental results show that the BC loss can be effective for tail/head evaluation and non-debiasing methods. Overall, the idea of using BC loss is interesting to me, and it also shows the theoretical analysis.

**Questions:**

- Q1) It is wondering whether the popularity bias effectively captures the popularity bias. Could you show some empirical or quantitative results?
- Q2) The source code does not work well. It may be related to pre-processing. Please check your code.


**Limitations:**

This paper does not address the negative societal impact. However, this paper seems not to have any negative impact.

**Strengths And Weaknesses:**

Strengths
- (Originality) The idea of using the popularity bias extractor is quite interesting.
- (Quality) The proposed loss outperforms existing methods in various evaluation protocols.
- (Clarity) It is well-written and easy to understand. Also, the theoretical proof is quite solid.


Weaknesses
- (Originality) Although the idea of incorporating the popularity bias into the contrastive loss is interesting, some recent studies also tried similar ideas. It is necessary that the proposed loss is compared with the following references.

[1] Kelong Mao, Jieming Zhu, Jinpeng Wang, Quanyu Dai, Zhenhua Dong, Xi Xiao, Xiuqiang He, SimpleX: A Simple and Strong Baseline for Collaborative Filtering, CIKM 2021

[2] Jiancan Wu, Xiang Wang, Xingyu Gao, Jiawei Chen, Hongcheng Fu, Tianyu Qiu, Xiangnan He, On the Effectiveness of Sampled Softmax Loss for Item Recommendation, 2022

- (Quality) Although the proposed model is compared with various debasing strategies, the models using inverse propensity scores (IPS) are not compared. It would be better to show that the proposed model is still better than some existing studies using IPS.
- (Quality) Although the proposed model shows the results on the KuaiRec dataset, most existing studies show two datasets for unbiased evaluation, i.e., Coat and Yahoo! R3. For a fair comparison, it would be better to show additional experimental results on these datasets.

---

> ### Author Response · Authors · 2022-08-02
> **Response to Reviewer jXGF - Part 1**
>
> We gratefully thank you for the valuable and constructive comments. To address your concerns, below we provide the point-to-point responses. We have carefully revised our paper by taking into account all your suggestions. Looking forward to more discussions with you.
>
> > **Comment 1: More experiments with other Losses** - “(Originality) Although the idea of incorporating the popularity bias into the contrastive loss is interesting, some recent studies also tried similar ideas. It is necessary that the proposed loss is compared with the following references. - [1] SimpleX; [2] SSM”
>
> Thanks so much for bringing these recent works to us. Following your suggestions, we have conducted additional experiments over different datasets to show the comparison between our BC loss, CCL adopted by SimpleX [1], and SSM [2]. The hyperparameters of $w$, $N$, $m$ in CCL and $\tau$ in SSM are carefully selected via grid search. We summarize the performance comparison in the following table. We have several observations: (1) Clearly, our BC loss consistently outperforms CCL and SSM; (2) CCL and SSM achieve comparable performance to BC loss in the imbalanced evaluation settings, while performing much worse than BC loss in the balanced evaluation settings. This indicates the superiority of BC loss in alleviating the popularity bias, and further justifies the effectiveness of the bias-aware margins.
>
> **Table 7 in Appendix: The performance comparison of loss functions on Tencent, Amazon, and iFashion**
> |             |   Tencent  |            |            |            |   Amazon   |            |            |            |  iFashion  |            |            |            |
> |-------------|:----------:|:----------:|:----------:|:----------:|:----------:|:----------:|:----------:|:----------:|:----------:|:----------:|:----------:|:----------:|
> |             |  Balanced  |            | Imbalanced |            |  Balanced  |            | Imbalanced |            |  Balanced  |            | Imbalanced |            |
> |             |   Recall   |    NDCG    |   Recall   |    NDCG    |   Recall   |    NDCG    |   Recall   |    NDCG    |   Recall   |    NDCG    |   Recall   |    NDCG    |
> | BPR         |   0.0052   |   0.0040   |   0.0982   |   0.0643   |   0.0109   |   0.0103   |   0.0850   |   0.0638   |   0.0056   |   0.0028   |   0.0843   |   0.0411   |
> | SSM         |   0.0055   |   0.0045   |   0.1297   |   0.0872   |   0.0156   |   0.0157   |   0.1125   |   0.0873   |   0.0079   |   0.0040   |   0.0963   |   0.0436   |
> | CCL         |   0.0057   |    00047   |   0.1216   |   0.0818   |   0.0175   |   0.0167   |   0.1162   |   0.0927   |   0.0075   |   0.0038   |   0.0954   |   0.0428   |
> | BC Loss | **0.0087** | **0.0068** | **0.1298** | **0.0904** | **0.0221** | **0.0202** | **0.1198** | **0.0948** | **0.0095** | **0.0048** | **0.0967** | **0.0487** |
> | Imp. %      |    52.6%   |    44.7%   |    0.1%    |    3.7%    |    26.3%   |    21.0%   |    3.1%    |    2.3%    |    20.3%   |    20.0%   |    0.4%    |    11.7%   |

---

> > ### Author Response · Authors · 2022-08-02
> > **Response to Reviewer jXGF - Part 2**
> >
> > > **Comment 2: Missing Baseline** - “(Quality) Although the proposed model is compared with various debasing strategies, the models using inverse propensity scores (IPS) are not compared. It would be better to show that the proposed model is still better than some existing studies using IPS.”
> >
> > Thanks. Actually, we used IPS-CN [3] as the representative baseline of IPS family. Specifically, it incorporates the idea of using propensity scores to reweight the samples, where the propensity weights are estimated via normalized capped importance sampling. As such, IPS-CN can resolve the high variance problem that frequently happened in IPS.
> >
> > [3] Offline evaluation to make decisions about playlist recommendation algorithms. 2019.
> >
> >
> > > **Comment 3: More experiments on Yahoo!R3 and Coat** - “(Quality) Although the proposed model shows the results on the KuaiRec dataset, most existing studies show two datasets for unbiased evaluation, i.e., Coat and Yahoo! R3. For a fair comparison, it would be better to show additional experimental results on these datasets.”
> >
> >
> > Thanks so much for your suggestions. We have conducted additional experiments on Coat and Yahoo! R3 datasets. Below we present the performance comparisons in the unbiased test sets. Clearly, our BC loss consistently achieves significant improvements over the baselines. This again indicates that BC loss truly improves the generalization ability of recommender.
> >
> > **Table 8: Performance comparison on Yahoo!R3 and Coat**
> > |         |  Yahoo!R3  |            |    Coat    |            |
> > |---------|:----------:|:----------:|:----------:|:----------:|
> > |         |   Recall   |    NDCG    |   Recall   |    NDCG    |
> > | IPS-CN  |   0.1081   |   0.0487   |   0.1700   |   0.1377   |
> > | CausE   |   0.1252   |   0.0537   |   0.2329   |   0.1635   |
> > | sam+reg |   0.1198   |   0.0548   |   0.2303   |   0.1869   |
> > | MACR    |   0.1243   |   0.0539   |   0.0798   |   0.0358   |
> > | BPR     |   0.1063   |   0.0476   |   0.0741   |   0.0361   |
> > | SSM     |   0.1470   |   0.0688   |   0.2022   |   0.1832   |
> > | CCL     |   0.1428   |   0.0676   |   0.2150   |   0.1885   |
> > | BC Loss | **0.1487** | **0.0706** | **0.2385** | **0.1969** |
> > | Imp. %  |    1.2%    |    2.6%    |    2.4%    |    4.5%    |
> >
> > > **Comment 4: Effectiveness of Popularity Bias Extractor** - “Q1) It is wondering whether the popularity bias effectively captures the popularity bias. Could you show some empirical or quantitative results?”
> >
> > We appreciate your suggestions. However, it is challenging to directly evaluate the popularity bias extractor, since no ground-truth interaction-wise bias is available. Hence, to address this concern, we have conducted additional experiments on the Tencent dataset and plotted three new figures in Appendix B.5.
> > - **Experimental Settings:** We first partitions interactions into disjoint subgroups, based on the bias degree estimated by our popularity bias extractor. These subgraphs are composed of (1) hard interactions with low bias degrees (40% of total interactions, Popularity rank > 1000), and (2) easy interactions with high bias degrees (15% of total interactions, Popularity rank < 100), respectively. Then, we use four losses (BPR, CCL, SSM, and BC losses) to train the same MF backbone.
> > - **Observations:** In Appendix B.5, Figures 5 depicts the performance of each loss in the hard-interaction subgroup, easy-interaction subgraph, and all-interaction group, during the training phase. With the increase in training epochs, the performance of BPR drops dramatically in the hard-interaction subgroup, while increasing in the all-interaction group. Possible reasons are BPR is easily influenced by the interaction-wise bias: BPR focuses largely on the majority subgroups of easy interactions, while sacrificing the performance on the minority subgroups of hard interactions. In contrast, the performance of BC loss consistently increases over every subgroup during training. This indicates that BC loss is able to mitigate the negative influences of popularity bias, thus further justifying that the popularity bias extract captures the interaction-wise bias well.
> >
> > > **Comment 5: Reproducibility** - ”Q2) The source code does not work well. It may be related to pre-processing. Please check your code.“
> >
> > Sorry for any inconvenience. We have updated the full version of codes with Tencent dataset. Please run the command line before running: python setup.py build_ext --inplace. Codes are available [here](https://anonymous.4open.science/r/BC-Loss-8764/model.py).

---

> > > ### Comment · Reviewer_jXGF · 2022-08-09
> > > **Additional questions**
> > >
> > >
> > > I have read all the reviews and the author feedback. Therefore, I raised my score.
> > >
> > > Additionally, I have some questions.
> > >
> > > Q1) I am still wondering if the popularity bias extractor works well. For instance, it would consider the ground-truth for each interaction bias as a user popularity bias + item popularity bias and compare it with the interaction-wise bias generated by the popularity bias extractor.
> > >
> > > Q2) Could you share detailed hyperparameter settings to reproduce the performance of the proposed BC loss on each dataset?

---

> > > > ### Author Response · Authors · 2022-08-09
> > > > **Follow-up your raised assessment**
> > > >
> > > > Dear Reviewer,
> > > >
> > > > May I check if we missed your raised score? Since your initial rating was 5, now it is still 5 ... Could we get your raised assessment?
> > > >
> > > > Thanks again for your additional questions! Our answers are on the way, and we will update them as soon as possible! Looking forward to further discussions with you!

---

> > > > ### Author Response · Authors · 2022-08-09
> > > > **Response to additional questions**
> > > >
> > > > We appreciate your time and effort.
> > > >
> > > > > Q1) I am still wondering if the popularity bias extractor works well. For instance, it would consider the ground-truth for each interaction bias as a user popularity bias + item popularity bias and compare it with the interaction-wise bias generated by the popularity bias extractor.
> > > >
> > > > Thanks so much for your great suggestions! However, we respectfully argue that using the sum of user and item popularity scores as the ground-truth of interaction-aware bias might be inappropriate, since there exists the scale gap between user and item popularity scores (e.g., the scores of users and items range in [0,70] and [0, 6000] on Tencent, respectively).
> > > >
> > > > To answer your question, we instead illustrate the relations between popularity scores and bias degree extracted by popularity bias extractor w.r.t. user and item sides, respectively. Specifically, Figure 7 in Appendix depicts their relationships, which are also quantitatively supported by Pearson correlation coefficients. (0.7703 and 0.662 for item and user sides, respectively). It shows the power of popularity embeddings to predict the popularity scores --- that is, user popularity embeddings derived from the popularity bias extractor are strongly correlated and sufficiently predictive to user popularity scores; analogously to the item side.
> > > >
> > > >
> > > >
> > > > > Q2) Could you share detailed hyperparameter settings to reproduce the performance of the proposed BC loss on each dataset?
> > > >
> > > > Please find the detailed hyperparameter settings in the following table.
> > > > |              | BC loss | hyper- | parameters |            |                      |
> > > > |--------------|:-------:|:------:|:----------:|:----------:|:--------------------:|
> > > > |              |    τ1   |   τ2   |     lr     | batch size | No. negative samples |
> > > > | **MF**       |         |        |            |            |                      |
> > > > | Tencent      |   0.06  |   0.1  |    1e-3    |    2048    |          128         |
> > > > | iFashion     |   0.08  |   0.1  |    1e-3    |    2048    |          128         |
> > > > | Amazon       |   0.08  |   0.1  |    1e-3    |    2048    |          128         |
> > > > | Douban       |   0.08  |   0.1  |    1e-3    |    2048    |          128         |
> > > > | Yahoo!R3     |   0.15  |   0.2  |    5e-4    |    1024    |          128         |
> > > > | Coat         |   0.09  |   0.4  |    5e-4    |    1024    |          64          |
> > > > | **LightGCN** |         |        |            |            |                      |
> > > > | Tencent      |   0.12  |   0.1  |    1e-3    |    2048    |       in-batch       |
> > > > | iFashion     |   0.14  |   0.1  |    1e-3    |    2048    |       in-batch       |
> > > > | Amazon       |   0.08  |   0.1  |    1e-3    |    2048    |       in-batch       |
> > > > | Douban       |   0.14  |   0.1  |    1e-3    |    2048    |       in-batch       |

---

> ### Author Response · Authors · 2022-08-08
> **Follow-up discussion**
>
> Thanks again for your insightful comments. Following your valuable suggestions, we have added two new datasets (Yahoo!R3 and Coat), three new baselines (BPR, CCL, SSM), and one new visualization experiment. We do hope our updated version can satisfy you. If we address your main concern, could you consider reassess our paper? If you have additional concerns, we would be pleased to discuss with you. We appreciate your time.

---

### Official Review · Reviewer_BZTL · 2022-07-20

**Rating:** 3
**Confidence:** 4
**Soundness:** 2 fair
**Presentation:** 3 good
**Contribution:** 3 good

**Summary:**

This paper proposes a loss function to mitigate the popularity bias in recommendation. It argues that existing debiasing methods poses two limitations: i) focusing on the trade-off between evaluations for popular and unpopular users/items (e.g. re-weighting popular and unpopular samples), ii) holding infeasible assumptions (e.g. accessibility to unbiased data). The proposed method can be divided into two parts: i) Evaluating the bias degree of each user-item pair, using the prediction error of a popularity-based interaction prediction network. ii) The proposed BC Loss, which is basically the softmax cross-entropy loss with the bias-specific margin based on the bias degree. The paper also gives geometric illustrations and some theoretical analysis on the strength of the proposed BC Loss.

**Questions:**

The essence of the proposed BC loss is to add popularity-related bias to the prediction scores of normal recommenders, which share the same debiasing paradigm with some existing debiasing works. One difference is that this paper measure the popularity bias using radius, which yields intuitive geometric interpretation.
The idea of employing an extra predictor based on popularity features to extract popularity degree is novel and promising. But the technical solution of this paper is rather simple, and the theoretical analysis does not give deep enough investigation on the re.

To evaluate the bias degree, if the popularity-based predictor achieves strong performance for a user-item pair, this pair is strongly influenced by the popularity.

**Limitations:**

Please refer to the identified weakness.

**Strengths And Weaknesses:**

Strength:
1.	The proposed model is a simple plugin suitable for various kind of recommendation methods.
2.	According to the experimental evaluations, the proposed BC loss indeed results in overall performance improvements for both popular and unpopular items.
Weakness:
1.	The technical contribution of this paper is limited. Comparing to existing CF methods, it only proposes to employ an extra popularity-based predictor and combine the results with an existing CF model.
2.	The paper overclaims the strength of the proposed BC loss in theoretical analysis. The geometric interpretability, theorem 1, the high/low entropy representations, and the hard-negative mining ability, are actually the same thing (i.e., applying stronger constrains for samples with higher popularity) from different viewpoints.

---

> ### Author Response · Authors · 2022-08-02
> **Response to Reviewer BZTL - Part 1**
>
> We appreciate your comments. To address your concerns, below we provide the point-to-point responses to prudently justify the novelty and clarify the misunderstandings of our proposed method. We have carefully revised our paper by taking into account all your suggestions. Looking forward to discussing more with you.
>
> > **Comment 1: Novel Idea, Simple Plugin, Performance Improvement** — “Strength: 1. The proposed model is a simple plugin suitable for various kind of recommendation methods. 2. According to the experimental evaluations, the proposed BC loss indeed results in overall performance improvements for both popular and unpopular items.” & “The idea of employing an extra predictor based on popularity features to extract popularity degree is novel and promising.”
>
> Thank you for the positive comments on the idea, applicability, and performance improvement of our proposed model. We would like to emphasize our main contribution: BC loss is a **simple yet effective** and **model-agnostic** strategy that incorporates popularity bias-aware margins into contrastive loss, so as to learn better representations for both heads and tails, instead of playing a trade-off game between heads and tails performance.
>
> > **Comment 2: Limited technical contribution** — “The technical contribution of this paper is limited. Compared to existing CF methods, it only proposes to employ an extra popularity-based predictor and combine the results with an existing CF model.”
>
> We respectfully argue that our BC loss is novel and significantly different from prior studies w.r.t. three technical aspects (See more details in Response 4’s Table, and we have clarified more about these aspects in the revision):
> - **Bias scope**: we consider popularity bias at the granularity of individual user-item interactions, while most prior studies focus only on either the user side or item side. Hence, jointly analyzing and formulating the interaction-wise biases from user and item sides is one of our technical contributions.
> - **Bias modeling**: we creatively use popularity embeddings to capture the bias degree. To the best of our knowledge, no works try to embed the bias. Hence, embedding popularity bias is another technical contribution.
> - **Debiasing Strategy**: we integrate the bias-aware margin into contrastive loss to guide the representation learning. This integration is strongly supported by geometric interpretation and information theory, rather than a “simple combination”. Specifically, (1) the prediction of popularity bias extractor is converted into the geometric angle between the user and item representations; (2) the angle is further transferred into the bias-aware angular margin, which is the key to discriminative power of representations, from the geometric perspective; and (3) from the perspective of information theory, the margin is integrated into the contrastive loss. To the best of our knowledge, no works debias from the geometric view and incorporate the bias into contrastive learning from the information theory.
>
> Moreover, we are encouraged that all the other reviewers (Reviewers jXGF and aXKZ) find our technical contributions novel and sufficient, rather than a “simple combination”. Specifically,
> - Reviewer jXGF: the BC loss “introduces a popularity bias extractor to estimate the popularity bias scores from a pair of user and item popularity. Then it incorporates it into the contrastive loss as the marginal information.”
> - Reviewer aXKZ: the BC loss “explored the debias problem from a geometric view and unified the bias term into a marginal contrastive learning framework elegantly.”
>
> Furthermore, the simplicity of BC loss does not mean the “limited technical contribution”, but indicates the easy use and wide applicability of BC loss. That is, it can be widely applied to a variety of recommender models and easily deployed in real-world scenarios, which can be conceptual advantageous over complicated designs. In a nutshell, we think our simple yet effective and model-agnostic BC loss holds sufficient technical contributions.

---

> > ### Author Response · Authors · 2022-08-02
> > **Response to Reviewer BZTL - Part 2**
> >
> > > **Comment 3: Theoretical Analysis of BC Loss** — “The paper overclaims the strength of the proposed BC loss in theoretical analysis. The geometric interpretability, theorem 1, the high/low entropy representations, and the hard negative mining ability, are actually the same thing (i.e., applying stronger constrains for samples with higher popularity) from different viewpoints.”
> >
> > Thanks for your comments. However, we respectfully argue that these theoretical analyses (geometric interpretation, Theorem 1, hard example mining mechanism) are describing different and unique properties of BC loss. We have clarified more about these analyses in the revision. Here we list their relations and differences:
> > - Geometric interpretation highlights different ranking assumptions implied by different losses (BPR, Softmax, and BC losses), which analyzes the posterior probabilities and illustrates these losses in both 2D and 3D hyperspheres. Hence, it showcases the advantage of BC loss w.r.t. **bias-aware decision boundary**, from the **geometric view**.
> > - Theorem 1 presents a lower bound of BC loss, which is a rigorous mathematical justification for why BC loss can learn better representations and enhance generalization ability. Hence, it supports the advantage of BC loss w.r.t. **representation learning**, from the **information-theoretical view**.
> > - Hard example mining mechanism (briefly analyzed in Appendix) shows BC loss’s ability to adaptively identify hard informative interactions with bias-aware margins, as compared to Softmax loss. Hence, it shows the advantage of BC loss w.r.t. adaptive mining over Softmax loss.
> >
> > Clearly, these aspects offer different, fresh, and unique insights into BC loss, rather than a coarse-grained and vague statement “applying stronger constraints for samples with higher popularity”. Hence, these aspects well justify the advantages of our BC loss.
> >
> > Moreover, we are also encouraged that all the other reviewers (Reviewers jXGF and aXKZ) find our theoretical analyses solid. Specifically,
> > - Reviewer jXGF: “The theoretical proof is quite solid.”
> > - Reviewer aXKZ: “The geometric interpretation and theoretical analysis also show the good properties of the proposed BC loss.”

---

> > > ### Author Response · Authors · 2022-08-02
> > > **Response to Reviewer BZTL - Part 3**
> > >
> > > > **Comment 4: Existing Debiasing Paradigm** — “The essence of the proposed BC loss is to add popularity-related bias to the prediction scores of normal recommenders, which share the same debiasing paradigm with some existing debiasing works.”
> > >
> > > Thank you for the comments. We respectfully argue two points:
> > > - Our BC loss does NOT “add popularity-related bias to the prediction scores of normal recommenders”. Specifically, it converts the interaction-wise popularity bias into the angular margin via Equation (6), and then achieves marginal contrastive learning.
> > > - Our BC loss is significantly distinct from the debiasing paradigm that some existing works follow, especially from the three aspects: (1) bias modeling, (2) debiasing strategy, and (3) evaluation. See Response 2 for the explanations w.r.t. the bias modeling and debiasing strategy. Below we summarize the detailed differences in the following table:
> > >
> > >
> > > |                 | **Bias Modelling**   |                                     |                            | **Debiasing Strategy**     |                    | **Evaluation**                                                                                        |                       |
> > > |:---------------:|----------------------|-------------------------------------|----------------------------|----------------------------|--------------------|-------------------------------------------------------------------------------------------------------|:---------------------:|
> > > |                 |    **Bias Scope**    |           **Measurement**           | **Space Interpretability** |      **Research Line**     |     **Theory**     |                                             **Data Split**                                            | **Test Distribution** |
> > > | **IPS-CN [1]**  |       Item-wise      | Normalized capped propensity weight |              -             |     Sample re-weighting    |          -         |                                            Online A/B test                                            |       Not known       |
> > > | **PC [2]**      |       Item-wise      |      Item popularity statistics     |              -             | Post-processing re-ranking |          -         |                                     In distribution randomly split                                    |       Not known       |
> > > | **Sam+reg [3]** |       Item-wise      |      Item popularity statistics     |              -             |    Regularization-based    |          -         |                                             Temporal split                                            |       Not known       |
> > > | **DICE [4]**    | User-wise; item-wise |   user/item popularity statistics   |              -             |          Causality         |          -         |                                   Out-of-distribution balanced split                                  |    Known in advance   |
> > > | **MACR [5]**    | User-wise; item-wise |     Predicted popularity scores     |              -             |          Causality         |          -         |                                   Out-of-distribution balanced split                                  |    Known in advance   |
> > > | **BC loss**     |   Interaction-wise   |         Popularity embedding        |          Geometric         |   Representation learning  | Information theory | Unbiased test set; in distribution randomly split; temporal split; Out-of-distribution balanced split |       Not known       |
> > >
> > >
> > > Reference:
> > >
> > > [1] Offline evaluation to make decisions about playlist recommendation algorithms. 2019.
> > >
> > > [2] Popularity-opportunity bias in collaborative filtering. 2021.
> > >
> > > [3] Connecting user and item perspectives in popularity debiasing for collaborative recommendation. 2021.
> > >
> > > [4] Disentangling user interest and conformity for recommendation with causal embedding. 2021.
> > >
> > > [5] Model-agnostic counterfactual reasoning for eliminating popularity bias in recommender system. 2021.

---

> ### Author Response · Authors · 2022-08-08
> **Follow-up discussion**
>
> Thanks again for your efforts in reviewing our paper. We hope our responses do address your misunderstandings, especially regarding the technical strength and theoretical properties. We thus do hope our clarification of this main concern does help you reassess our paper. If you have additional concerns, we'd be more than happy to provide additional clarification. Thank you for your attention.

---

> ### Author Response · Authors · 2022-08-09
> **A Gentle Reminder for the End of Rebuttal Period**
>
> We thank the reviewer again for your comments. This is a gentle reminder for the end of the discussion session. We have updated our submission and posted point-to-point responses to your comments. We would be grateful if you could confirm whether our responses have addressed your concerns. We hope that the rebuttal will be taken into consideration.

---

### Author Response · Authors · 2022-08-04
**Response to reviewers**

We appreciate all the reviewers for their valuable comments and suggestions. This helped improve our submission and better strength our claims. Here we summarize the major updates brought to the revised manuscript:

- **More baselines with experiments**. Addressing a shared concern of Reviewers jXGF and aXKZ, we have added a comparison between BC loss and baselines of standard loss functions (e.g., BPR, CCL, SMM) in collaborative filtering in Appendix B.4.

- **More datasets with experiments**. Following the suggestions of Reviewer jXGF, we have conducted additional experiments on Coat and Yahoo! R3 datasets and reported the results in Appendix B.3.

- **More visualization**. To better support effectiveness of our popularity bias extractor, we have added: 1). visualization of bias angle in different interaction types, in response to Reviewer aXKZ's comments; and 2) additional experiments to visualize the performance of each loss in different types of interactions to address the concern of Reviewer jXGF.

We have tried our best to address the main concerns raised by reviewers and we hope that these improvements will be taken into consideration. Updates in the revision are highlighted in blue. We also present the point-to-point responses for each reviewer below.

---

### Author Response · Authors · 2022-08-08
**Ask for your help**

Dear area chairs:

We would like to bring up your attention to our paper. The majority of reviewers' concerns regarding to our paper, in our opinion, are related to inclusion of more experiments. In the rebuttal period, we have thoughtfully provided the point-to-point responses raised by the reviewers. The revision has added a large amount of additional experiments in response to reviewers' comments. Even though it has taken a lot of our time and effort to resolve every reviewer's concern, no discussion appears ...

The deadline for author-reviewer discussion is approaching. We sincerely hope that our effort and improvement can be taken into consideration. We would like to ask could you please help us to remind the reviewers? Thanks again.

---

### Meta-Review · Area_Chair_juMT · 2022-08-26

**Recommendation:** Accept
**Confidence:** Less certain

**Metareview:**

This paper studies the popularity bias of collaborative filtering-based recommendation systems. Specifically, this paper proposes a bias-aware margin into the contrastive loss, resulting in a modified BC loss to remedy the problem. Geometric interpretation and experimental results are provided to validate the effectiveness of the proposed method. This paper received mixed review comments. A review raises concerns about the technical novelty and contribution of the proposed method mainly due to the simplicity of the proposed solution. Considering that the simple solution is well supported by theoretical justification and empirical results, I lean to accept this paper.


**Award:**

No

---

### Decision · Program_Chairs · 2022-09-14

Accept